# Timing of gene expression in a cell-fate decision system

Delphine Aymoz[1] iD, Carme Solé[2] iD, Jean-Jerrold Pierre[1] iD, Marta Schmitt[1], Eulàlia de Nadal[2] iD, Francesc Posas[2] iD & Serge Pelet[1],* iD

## Abstract

During development, morphogens provide extracellular cues allowing cells to select a specific fate by inducing complex transcriptional programs. The mating pathway in budding yeast offers simplified settings to understand this process. Pheromone secreted by the mating partner triggers the activity of a MAPK pathway, which results in the expression of hundreds of genes. Using a dynamic expression reporter, we quantified the kinetics of gene expression in single cells upon exogenous pheromone stimulation and in the physiological context of mating. In both conditions, we observed striking differences in the timing of induction of mating-responsive promoters. Biochemical analyses and generation of synthetic promoter variants demonstrated how the interplay between transcription factor binding and nucleosomes contributes to determine the kinetics of transcription in a simplified cell-fate decision system.

**Keywords** gene expression; MAPK pathway; single-cell measurements; yeast mating

**Subject Categories** Quantitative Biology & Dynamical Systems; Signal Transduction; Transcription

**Mol Syst Biol. (2018) 14: e8024**

## Introduction

Cell-fate decisions play a key role in crucial processes such as tissue repair, immune response, or embryonic development. In order to make choices, cells integrate cues from neighboring cells as well as from morphogens. Signal transduction cascades relay this information inside the cell to translate these extracellular signals into defined biological responses. The cellular output includes the induction of complex transcriptional programs where specific genes are expressed to different levels and at various times (Gurdon *et al*, 1995; Ashe *et al*, 2000). Ultimately, these different expression programs will determine the fate of individual cells. The mating pathway in budding yeast has often been considered as a simplified

cell-fate decision system, where each cell can either continue to cycle in the haploid state or decide to mate with a neighboring cell of opposing mating type. This decision results in an arrest of the cell cycle and formation of a mating projection and ultimately leads to the fusion with the partner to form a diploid zygote (Bardwell, 2005; Atay & Skotheim, 2017).

Haploid budding yeast senses the presence of potential mating partners by detecting pheromone in the medium. This small peptide elicits the activation of a mitogen-activated protein kinase (MAPK) cascade (Appendix Fig S1), which can integrate multiple cues such as stresses, cell cycle stage, or nutrient inputs (Strickfaden *et al*, 2007; Doncic *et al*, 2011; Nagiec & Dohlman, 2012; Clement *et al*, 2013). Once the MAPKs Fus3 and Kss1 are activated, they phosphorylate a large number of substrates and induce a new transcriptional program. Ste12 is the major transcription factor (TF) implicated in this response and controls the induction of more than 200 genes (Roberts *et al*, 2000). Under normal growth conditions, this TF is repressed by Dig1 and Dig2. Phosphorylation by active Fus3 and Kss1 relieves this inhibition, such that Ste12 can recruit the transcriptional machinery (Tedford *et al*, 1997). Ste12 associates with the DNA via well-established binding sites located in promoters called pheromone response elements (PRE), with the consensus sequence ATGAAACA (Kronstad *et al*, 1987; Hagen *et al*, 1991). Although PREs are found upstream of the vast majority of pheromone-induced genes (Chou *et al*, 2006), the number of binding sites, their orientation, and their position relative to the transcription start site vary widely from one gene to the next (Chou *et al*, 2006; Su *et al*, 2010).

Promoter sequences are primary determinants of the strength and kinetics of gene expression. Unfortunately, the basic rules governing transcription regulation remain poorly understood. Libraries of synthetic promoter sequences have allowed establishing a few rules in the control of the expression level and the noise of a promoter sequence (Sharon *et al*, 2012; Levo & Segal, 2014; Hansen & O'Shea, 2015). However, the slow maturation time of fluorescent proteins (FP) precluded thorough investigations of gene expression kinetics. In a previous paper, we developed the dPSTR, a fluorescent relocation reporter that converts the expression of a promoter into a signal of relocation of a fluorescent protein (Aymoz *et al*, 2016).

1 Department of Fundamental Microbiology, University of Lausanne, Lausanne, Switzerland
2 Cell Signaling Research Group, Departament de Ciències Experimentals i de la Salut, Universitat Pompeu Fabra, Barcelona, Spain
*Corresponding author. Tel: +41 21 692 5621; E-mail: serge.pelet@unil.ch

In this study, we use these dynamic gene expression reporters to characterize the induction dynamics of a set of promoters activated in response to yeast mating pheromone. We have identified different classes of promoters based on the kinetics of their expression. Deeper analysis of early and late promoters highlighted the interplay between TF binding and nucleosome positioning as a major determinant of the expression dynamics. In addition, we demonstrate that under physiological mating conditions, the induction of the target genes follows a precise chronology and they are sequentially expressed until fusion occurs.

## Results

### Interplay between kinase activity and expression dynamics

In multiple MAPK pathways, MAPK activity has been shown to be tightly linked to the transcriptional process by phosphorylating TFs, contributing to the recruitment of remodeling complexes, and participating in the elongation complex (de Nadal *et al*, 2011). Therefore, we wanted to measure, in the mating pathway, how kinase activity and gene expression were temporally correlated. Using fluorescent relocation sensors that we previously engineered, we are able to quantify, in real-time and at the single-cell level, both MAPK activity and gene expression upon stimulation of *MATa* cells with synthetic pheromone ($\alpha$-factor, 1 $\mu$M; Durandau *et al*, 2015; Aymoz *et al*, 2016). Signaling activity was quantified using a Ste7$_{DS}$-SKARS$^Y$, which exits the nucleus when the mating MAPKs Fus3 and Kss1 phosphorylate specific residues in the vicinity of a nuclear localization sequence (NLS) (Fig 1A; Appendix Fig S2A). In the same cells, a dynamic protein expression reporter p*FIG1*-dPSTR$^R$ was integrated. *FIG1* displays the largest fold induction upon pheromone stimulation (Roberts *et al*, 2000). In this assay, the *FIG1* promoter drives the expression of a small peptide, which interacts with a fluorescent protein and promotes its recruitment in the nucleus (Fig 1A, Appendix Fig S2B, Aymoz *et al*, 2016). Upon stimulation, the cells activate the mating MAPKs a few minutes after stimulation, as previously described (Yu *et al*, 2008; Nagiec & Dohlman, 2012; Durandau *et al*, 2015). Despite this fast signal transduction, the resulting p*FIG1* expression occurs 30 min later (Fig 1A and C). Individual yeast cells are known to possess a large diversity in signaling capacity (Colman-Lerner *et al*, 2005; Strickfaden *et al*, 2007). However, the expression dynamics of p*FIG1* is still highly variable

within the sub-population of cells that activate the MAPK within the 10 min following stimulation, suggesting that the heterogeneity in p*FIG1* expression does not result from various kinetics of MAPK activation (Appendix Fig S3A). This finding suggests an absence of temporal correlation between kinase activity and the downstream transcriptional response.

This surprising result led us to test the expression kinetics of multiple mating-responsive promoters. Among them was *AGA1*, a gene reported to be strongly induced upon pheromone stimulation (Roberts *et al*, 2000; McCullagh *et al*, 2010). The p*AGA1*-dPSTR$^R$ begins to enrich in the nucleus of cells 15 min after stimulation (Fig 1B and D). Thus, the induction of gene expression from this promoter is much faster than for p*FIG1*. In addition, the induction of p*AGA1* in signaling-competent cells is less variable with the vast majority of the cells inducing the reporter within 30 min following the stimulus (Appendix Fig S3A). This raises the question of how the activation of these two promoters is related in a same cell.

### Direct comparison of two dynamic expression reporters

We used a second protein expression reporter, the dPSTR$^Y$, which is orthogonal to the dPSTR$^R$, to quantify p*AGA1* and p*FIG1* expression dynamics in the same strain (Aymoz, *et al*, 2016; Fig 1E and Appendix Fig S3B). In all expressing cells, the response time for each promoter was determined based on the time at which the dPSTR nuclear enrichment reached 20% of its maximum (Fig 1F, see Materials and Methods and Appendix Fig S4). p*AGA1* expression is relatively homogeneous between cells, with 83% of the cells inducing the promoter within the first 30 min following stimulation. In comparison, p*FIG1* expression is highly variable from cell to cell. In cells inducing both promoters, the difference in response times can be measured (Fig 1F, inset). In 87% of cells, the p*AGA1*-dPTSR$^Y$ is activated prior to the p*FIG1*-dPSTR$^R$, which on average is delayed by 23 min. These different dynamics of induction are also well illustrated by the absence of correlation between the dPSTR enrichment seen at early time points (Fig 1G). The cell population becomes first p*AGA1* expressing, as denoted by a shift along the *x*-axis. Later, a shift of the cell population is observed along the *y*-axis, illustrating the delay in the induction of p*FIG1*. This delay is not an artifact from the dPSTRs, since the same results can be obtained when exchanging the promoters on the dPSTRs (Appendix Figs S5 and S6). In parallel, we have also verified that mRNA production dynamics

**Figure 1. Interplay between kinase activity and promoter induction in the mating pathway.**

A, B   Microscopy images of cells stimulated with a saturating pheromone concentration (1 $\mu$M) at time 0 min. The cells bear a histone tagged with CFP, a yellow SKARS reporting on Fus3p and Kss1p activities, and a red dPSTR reporting on p*FIG1* (A) or p*AGA1* (B) induction. For all experiments, unless stated otherwise, the stimulation was performed by addition of 1 $\mu$M $\alpha$-factor at time 0 min.

C, D   Quantifications of the kinase activity (green, left axis), measured by the ratio of cytoplasmic to nuclear YFP, and of the p*FIG1* (C) and p*AGA1* (D) expressions, measured by the difference between nuclear and cytoplasmic fluorescence of the dPSTR (right axis). For all similar graphs, the solid line is the median response and the shaded area represents the 25th–75th percentiles of the population.

E       Microscopy images of a strain carrying p*FIG1*-dPSTR$^R$ and p*AGA1*-dPSTR$^Y$.

F       Quantification of the response time of p*FIG1* and p*AGA1* reporters (see Materials and Methods). The inset is the difference response time between the p*AGA1*-dPSTR$^Y$ and the p*FIG1*-dPSTR$^R$, for all cells expressing both promoters. The red shaded area represents cells expressing p*AGA1* before p*FIG1* (87%).

G       Correlation of normalized dPSTR nuclear enrichments from all single cells of a representative experiment at different time points after stimulation.

H       Northern blot detection of mRNAs from *AGA1* and *FIG1* after stimulation of the cells with mating pheromone. See also Appendix Fig S15.

Data information: All scale bars on microscopy images represent 2.5 $\mu$m.

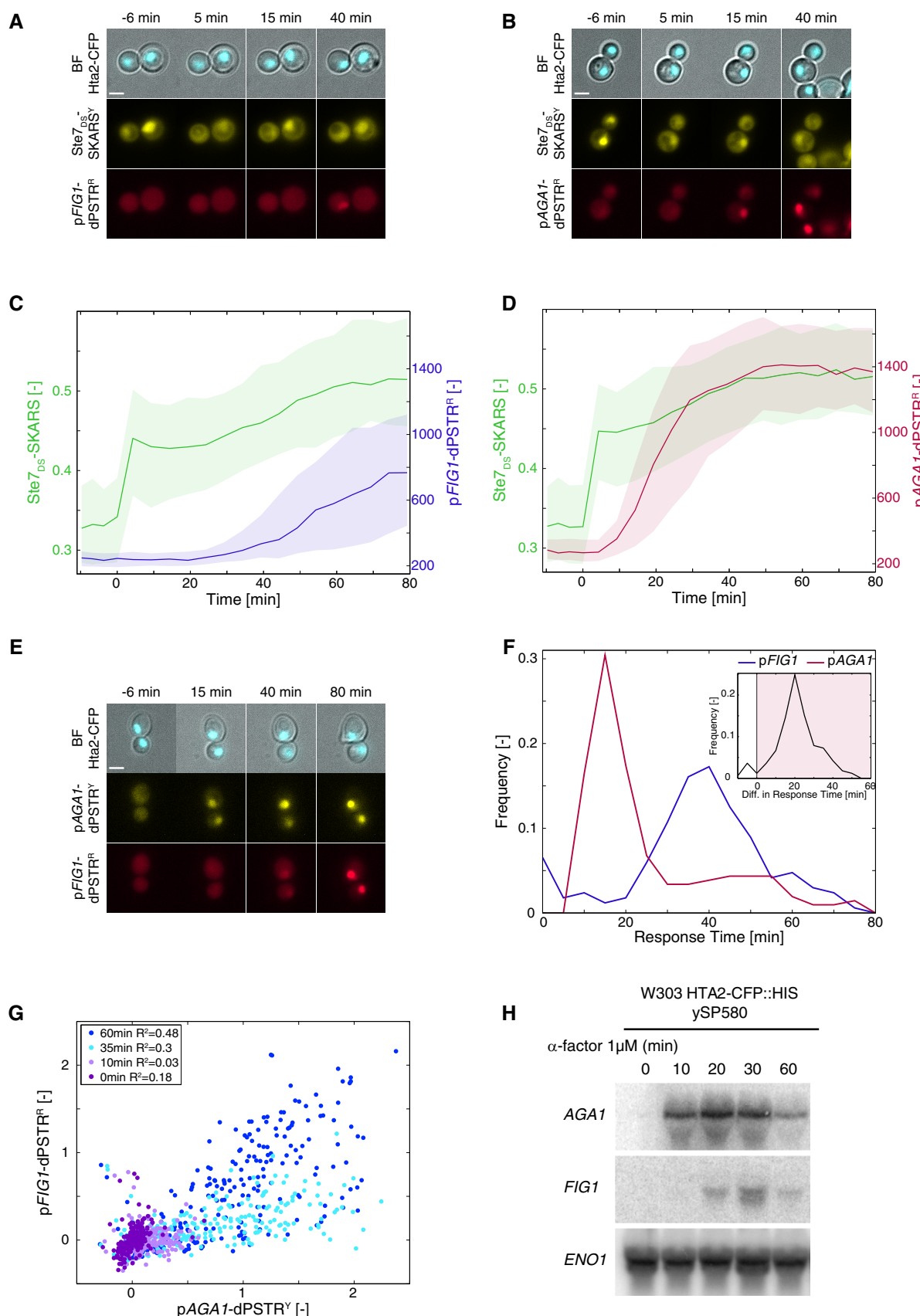

**Figure 1.**

from these two promoters correlate well with the expression dynamics we measured with the dPSTR (Fig 1H, Appendix Fig S7). Together, these data demonstrate that although the MAPK activity rises quickly in response to pheromone sensing, it does not lead to a fast and simultaneous transcriptional activation of all mating genes.

### Characterization of mating-induced promoters

Having established that the two promoters pAGA1 and pFIG1 are induced with different kinetics following pheromone stimulation, we tested when other mating-induced genes were induced with respect to pAGA1. Fourteen mating-responsive promoters, previously described in the literature, were characterized using a dPSTR[R] (Fig EV1; Roberts *et al*, 2000; Chou *et al*, 2006; Su *et al*, 2010). We quantified for each of them their expression output, by measuring the maximal variation of the nuclear enrichment of the dPSTR[R] upon stimulation (see Materials and Methods, Appendix Fig S4). These promoters display a large variability both in the level of induction and in the timing of expression (Fig 2A, Dataset EV1). Some genes are expressed early as *AGA1* (*FUS1*, *FAR1*, *STE12*, etc...); others are late responders similar to *FIG1* (*PRM3*, *KAR3*).

In order to better characterize the dynamics of expression of the 14 promoters, they were compared to the same internal control, a pAGA1-dPSTR[Y]. The difference in response time relative to pAGA1 induction was calculated (Figs 2B and EV2). In addition, the comparison of the overall dynamics of induction was visualized by plotting the mean nuclear enrichment of the yellow and red dPSTRs, normalized between their basal and maximal expression levels (Figs 2C and EV2). Each curve represents the correlation of the normalized expression levels of the two measured promoters and its evolution in course of the time-lapse, going from the bottom left to the upper right corner. Promoters which are induced with similar dynamics as pAGA1 will remain close to the $x = y$ diagonal (dashed line). Any difference in induction dynamics will cause a deviation from this line. Based on these measurements, we defined three classes of promoters: early, intermediate, and late. The early promoters, with kinetics similar to pAGA1, display a difference in response time centered around zero and a correlation aligned on the $x = y$ diagonal (Fig EV2). Late promoters, which behave similarly to pFIG1, have a response time delayed by at least 15 min and a correlation strongly deviating from the diagonal. Between these two clearly identifiable groups, a set of promoters display intermediate kinetics, where the response time is slightly delayed and/or where the dynamic correlation with pAGA1 is significantly deviating from the pAGA1/pAGA1 correlation at many time points.

The basal level of expression before stimulus (Appendix Fig S8) or the maximal expression level reached after pheromone induction (Fig 2A) does not allow to predict whether a promoter will be fast or slow. For instance, the *STE12* promoter belongs to the early genes group, but possesses one of the lowest induction levels. However, there is a clear link between the ability to respond at low pheromone concentration and the dynamics of promoter induction (Fig EV3, Appendix Fig S9). pAGA1 and other promoters from this category display a graded response as α-factor concentration increases. In comparison, late promoters behave in a more switch-like manner (Hill coefficient close to 3), where gene expression occurs only at high concentrations of α-factor (300 nM).

### Variability in gene expression

When focusing on the single-cell responses, a remarkable correlation between the expressions of the fast promoters at various time points can be observed (pAGA1/pFUS1: Fig 2D and other pairs in Appendix Figs S10 and S11). This tight correlation can be explained by the low noise present in the mating pathway and the expression variability being mostly governed by extrinsic variables such as the cell cycle stage and the expression capacity (Colman-Lerner *et al*, 2005). More striking is the fact that two late promoters in the same cell are also induced with a good correlation. This implies that despite the fact that the induction of these late genes can occur from 30 to 80 min after the stimulus, these two promoters are activated synchronously within a given cell (Fig 2E and Appendix Figs S6 and S11). These data also allow to rule out the presence of a slow stochastic activation of the late genes and rather argue in favor of a specific commitment point that the cells reach when they start to induce the late promoters.

In order to illustrate this better, we defined the correlative promoter variability (CPV), which allows to quantify the deviation in the induction of two promoters measured in the same cell, relative to the overall noise in expression (Fig 2F, Appendix Fig S6 and Fig EV2, see Materials and Methods). For two promoters well correlated like pAGA1 and pFUS1 (Oehlen *et al*, 1996), the CPV starts below 50% and tends to further decrease upon pheromone-dependent induction. Among fast promoters, there can be different types of behavior, depending mostly on the pre-stimulus levels of the reporter. The variability between pFAR1 and pAGA1 is a good illustration of this (Fig EV2). The CPV is high in basal conditions, because pAGA1 and pFAR1 are both transiently expressed during the cell cycle, although in different phases (Appendix Figs S8 and S10, Oehlen *et al*, 1996). However, following stimulation with pheromone, the variability decreases quickly as the two promoters are simultaneously induced. In comparison, the CPV between the late *FIG1* promoter and the early pAGA1 increases during the first 20 min following induction, due to an asynchronous induction of pAGA1 and pFIG1. Upon activation of the late promoter, the variability decreases. The CPV value comparing the two slow promoters pFIG1 and pKAR3 is around 60% before stimulation (Fig 2F, blue curve). This value is higher than 50% because of the cell cycle driven induction of pKAR3, leading to various basal levels of this promoter, whereas pFIG1 is not expressed in absence of stimulation (Kurihara *et al*, 1996; Appendix Fig S8). After stimulus, this CPV level is maintained for roughly 30 min, during which none of these two promoters are induced and then drops. Overall, these measurements demonstrate that each mating-induced promoter is expressed with specific dynamics and expression level. Some cells will induce the early genes few minutes after the stimulus, while in the same cell, other genes can be expressed up to 40 min after the first wave of gene expression. Remarkably, the tight co-regulation of early and late genes within their group strongly suggests that a shared mechanism exists that regulates the early promoters, which is different from the one controlling the activation of the late promoters.

### Architecture of mating promoters

In order to understand how the timing of induction is regulated, we have mapped all putative Ste12 binding sites in the sequences of the

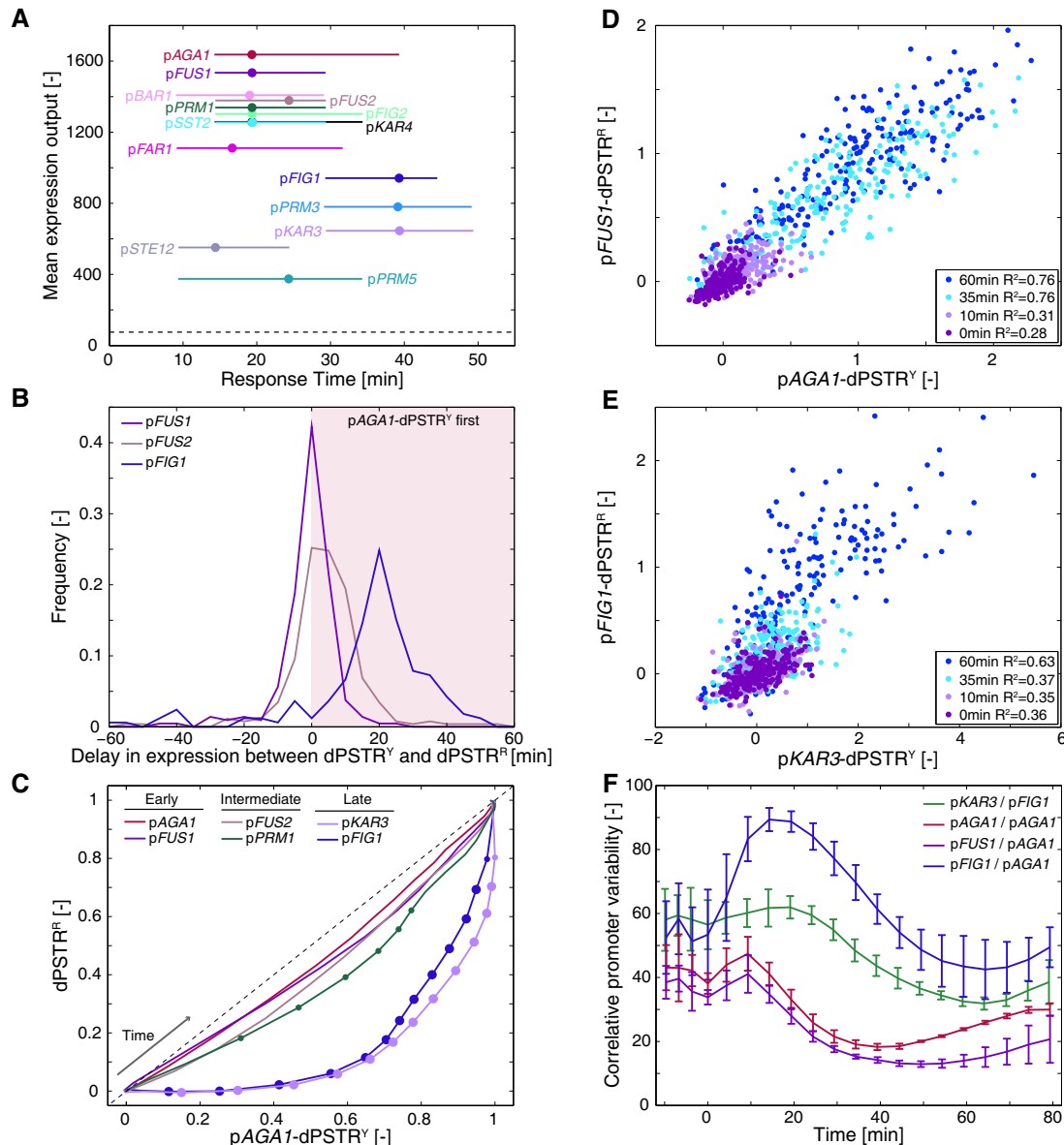

**Figure 2. Dynamics of induction of mating promoters after pheromone stimulation.**

A    Response time versus mean expression output for the 14 mating-dependent promoters. Dots represent the median response times of the cell population, and lines represent the 25th and 75th percentiles. All promoters were measured with the dPSTR$^R$. The strains also bear the p*AGA1*-dPSTR$^Y$ for direct comparison of the dynamics of promoter induction. The dashed line represents the detection sensitivity of the dPSTR$^R$ reporter.

B    Distributions of the differences in the response times between the p*AGA1*-dPSTR$^Y$ and the dPSTR$^R$ in the same cell for p*FUS1*, p*FUS2*, and p*FIG1*.

C    Correlation of the population-averaged normalized nuclear enrichment of p*AGA1*-dPSTR$^Y$ and a selected set of promoters measured with the dPSTR$^R$ at all time points of the experiments. The curves show the evolution in course of the experiment, from the bottom left to upper right corner, of the expression levels of the two measured promoters. The dots represent the $P$-value ($10^{-3} > P > 10^{-6}$ for small dots and $P < 10^{-6}$ for large dots) of the $t$-test comparing the offset of the measured promoter relative to the $x = y$ line with the offset of the reference promoter p*AGA1*.

D, E    Correlation of normalized dPSTR nuclear enrichments of single cells of at different time points after stimulation in a strain with p*FUS1*-dPSTR$^R$ and p*AGA1*-dPSTR$^Y$ (D) or p*FIG1*-dPSTR$^R$ and p*KAR3*-dPSTR$^Y$ (E).

F    Evolution of the correlative promoter variability (CPV) in course of time, for various pairs of promoters. The curve represents the mean of three replicates, and the error bar represents the standard deviation between replicates. A low CPV corresponds to a similar expression between two promoters in the same cell (see Materials and Methods).

fourteen promoters (Appendix Fig S12). We defined consensus PREs as nTGAAACn, as it was reported that these six core nucleotides were the most important to promote Ste12 binding *in vitro* (Su *et al*,

2010). We also identified several non-consensus PREs that carry additional mutations within the six core nucleotides. These putative binding sites possess a decreased affinity for Ste12, but can

contribute to Ste12-mediated expression (Su *et al*, 2010). As reported previously, there is a large variability in the number, orientation, spacing, and sequences of PREs among all promoters (Chou *et al*, 2006; Su *et al*, 2010). Therefore, there is no obvious rule that would allow to predict whether a gene is early- or late-induced, or expressed at low or high levels. Interestingly, p*AGA1* and p*FIG1* possess three consensus PREs with relatively similar dispositions and orientations and respectively four and five non-consensus PREs (Fig 3A and B). Despite these similarities, we have observed drastic differences in their expression kinetics. Therefore, we decided to use p*AGA1* and p*FIG1* as model promoters of their categories and decipher their mode of regulation.

### Regulation of *pAGA1* and *pFIG1*

In a strain bearing the p*FIG1*-dPSTR[R] and the p*AGA1*-dPSTR[Y] reporters, key regulators of the pathway were deleted. A number of mutants did not affect the expression from both promoters (group I: *kss1*Δ, *mot3*Δ, *tec1*Δ, *dig2*Δ, *bar1*Δ, and *arp8*Δ; Appendix Fig S13) or altered it in a similar fashion (group II: *ste12*Δ, *ste2*Δ, *ste11*Δ, *dig1*Δ, and *dig1*Δ*dig2*Δ; Appendix Fig S14). These mutants provide the anticipated phenotype except for the *dig1*Δ strain where a delay in gene expression is observed for both p*AGA1* and p*FIG1*. We hypothesize that the strong derepression of filamentous genes in this mutant (Chou *et al*, 2006) might limit the number of available Ste12 molecules needed to activate rapidly the mating-dependent promoters.

However, the most interesting knockouts are the ones that perturbed one promoter to a greater extent than the other one (group III, Fig EV4). In *fus3*Δ and *far1*Δ cells, p*AGA1* induction is delayed while p*FIG1* is severely reduced. Only a small percentage of cells induce p*FIG1*. Cells deleted for a member of the SAGA chromatin remodeling complex (*gcn5*Δ) also displayed a stronger decrease in p*FIG1* induction than in p*AGA1*, suggesting a higher requirement for nucleosome modification at the *FIG1* than at the *AGA1* promoter. Finally, deletion of the transcription factor *KAR4* profoundly affects p*FIG1* induction, without noticeable changes in p*AGA1*-dPSTR[Y] expression. Kar4 has been identified as a transcription factor required for the induction of genes implicated in karyogamy, a late event of the mating (Kurihara *et al*, 1996). Microarray measurements have identified a set of genes, such as *KAR3* and *PRM3*, that depend on Kar4, but *FIG1* was not one of them (Lahav *et al*, 2007). It has also been suggested that Kar4 forms a heterodimer with Ste12, and therefore, the association of those two proteins on the promoter allows the transcription of the late genes (Lahav *et al*, 2007). Moreover, we found that *KAR4* is induced as early as p*AGA1* during the mating response, making it a good candidate to regulate late genes.

### Ste12 and Kar4 interplay at the promoter

In order to better understand the sequence of events taking place at these two promoters, we monitored transcription factor binding by chromatin-IP, chromatin remodeling by MNase assays, and mRNA production by Northern blot. All these experiments were performed in the same strain with Ste12-myc and Kar4-HA tags. We noticed that the presence of these tags slightly influences the dynamics of transcription although the differential response of the two promoters

is maintained (Appendix Fig S15A and B). On the *AGA1* promoter, a fast enrichment of Ste12 and Kar4 is observed within 5 min after stimulus. In parallel, the chromatin is remodeled on the locus, as visualized by the eviction of the −1 nucleosome (Fig 3C, Appendix Fig S15C and Dataset EV2). The concomitant enrichment in TF and opening of the chromatin result in a rapid production of mRNA. In comparison, at the *FIG1* locus, all these events happen more slowly (Fig 3D and Appendix Fig S15D). Ste12 and Kar4 reach a maximal accumulation at 30 min after the stimulus, a time point where chromatin remodeling starts to take place. As a consequence, the resulting mRNA production is delayed at this locus.

The ability of TFs to bind promoter regions is known to depend on the positioning of nucleosomes on the DNA. MNase protection assays, in agreement with genome-wide studies (Appendix Fig S15C and D; Brogaard *et al*, 2012), allow to predict which PRE could be accessible under basal conditions. On p*AGA1*, two consensus binding sites for Ste12 are present in a nucleosome-depleted region (Fig 3A). This conformation would allow the formation of a Ste12 dimer under basal conditions. Indeed, both Ste12 and Kar4 are found associated with *AGA1* and *FIG1* promoters even before the addition of α-factor (Appendix Fig S15E). The Ste12 dimer on p*AGA1* could allow a fast induction of transcription as soon as Fus3 activity is present to derepress Dig1 and Dig2. In agreement with this prediction, mutation of either of these PRE sites delays significantly the induction of p*AGA1* transcription, and mutation of both PREs virtually abolishes the induction of this promoter variant (Appendix Fig S16A–D).

In comparison, only one strong Ste12 binding site is found in a nucleosome-depleted region of p*FIG1* (Fig 3B). This site lies in the close vicinity of a non-consensus site. Surprisingly, mutation of either of these two sites completely abolishes the mating-dependent induction from these promoter variants (Appendix Fig S16E–H). Note that deletion of either of the two other consensus PREs of p*FIG1* only lead to a mild defect in expression. In order to understand the parameters that control the dynamics of induction of the late promoters, we performed a series of mutations to test whether we succeeded to accelerate the dynamics of induction of the p*FIG1* promoter. In a first variant, we mutated the non-consensus site of p*FIG1* into a consensus one. This operation could putatively allow the recruitment of a Ste12 dimer under basal conditions, because both binding sites fall in a nucleosome-depleted region of the *FIG1* locus. This promoter variant was only marginally faster than the WT promoter. However, this single point mutation in the non-consensus site renders the induction of this promoter Kar4-independent (Appendix Fig S16I and J).

To alter the nucleosome landscape on p*FIG1*, we constructed a promoter chimera and replaced the 150 bp of the core promoter that is associated with −1 nucleosome in p*FIG1* by the p*AGA1* sequence (Fig 3E green curves, Appendix Fig S16I and J). This promoter chimera displays an intermediate behavior between p*FIG1* and p*AGA1*. It is faster and more expressed than the natural p*FIG1* promoter and retains a Kar4 dependency. One hypothesis is that the affinity of the −1 nucleosome for DNA is encoded in this sequence. Bringing this more labile nucleosome on the p*FIG1* promoter accelerates the expression of this construct. By combining the two modifications (non-consensus to consensus PRE in the chimera), we further accelerated the induction of the promoter and rendered it Kar4-independent (Fig 3E pink curves, Appendix Fig S16I and J).

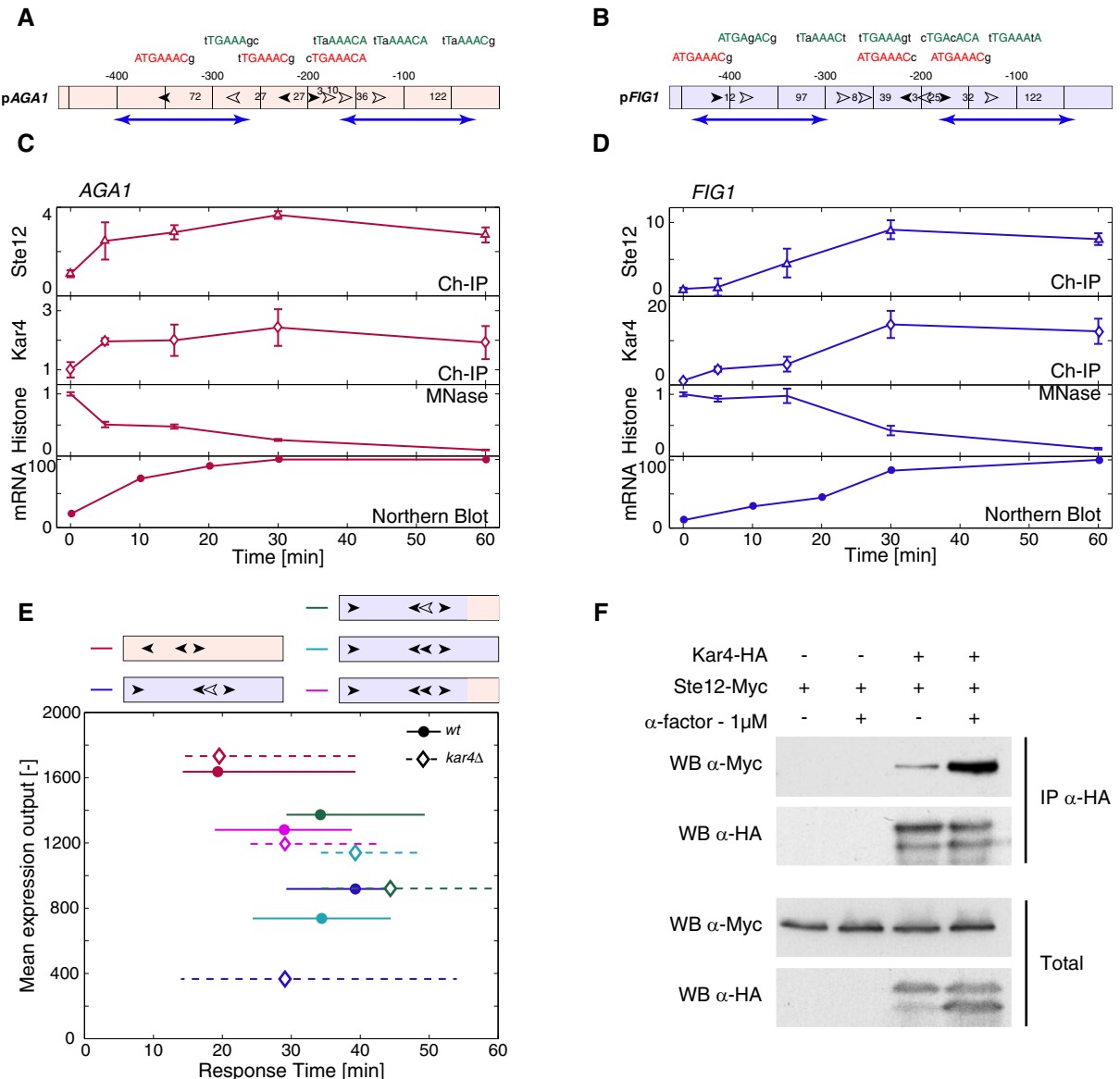

**Figure 3.   Influence of promoter architecture on expression dynamics.**

A, B   Maps of the two promoters pAGA1 and pFIG1. The filled arrows represent the location and orientation of consensus Ste12-binding sites (nTGAAACn). The open arrows symbolize the non-consensus binding sites that possess mutations within the six core nucleotides of the PREs. The sequences of each binding sites are detailed above, with capital nucleotides matching the consensus sequences and small nucleotides being mutations from the consensus. The numbers between sites represent the distance in bp between them or the ATG. Blue arrows represent nucleosomes position (Brogaard *et al*, 2012).

C, D   Quantification of molecular events at the *AGA1* (C) and *FIG1* (D) loci. Fold increase in Ste12-myc and Kar4-HA binding at the promoter quantified by chromatin-IP (open markers). Normalized −1 histone occupancy quantified by micrococcal nuclease (MNase) digestion. Transcript levels of *AGA1* (C) and *FIG1* (D) quantified by Northern blot (rounds). Each data point is the mean of three biological replicates, and the error bars represent their standard deviation.

E   Response time versus mean expression output for various promoters in a WT background (circles, solid lines) or *kar4Δ* (diamonds, dashed lines) background, as described in Fig 2A. Red is pAGA1, blue is pFIG1, green is a chimeric construct between pFIG1 and the last 150 bp of pAGA1, cyan is a construct where the free non-consensus binding site of pFIG1 (−209) was mutated into a consensus one, and purple is a combination of the chimeric construct with the mutation of the non-consensus binding site into a PRE.

F   *In vivo* binding of Ste12 and Kar4 was assessed by immunoprecipitation of Kar4p-HA and detection of Ste12-Myc in the presence and absence of pheromone.

The combination of the fast Ste12 dimer formation with the possible displacement of the nucleosome −1 favors a fast induction of this promoter, which no longer requires Kar4 presence.

Taken together, these data allow us to infer a model where early genes possess at least two consensus binding sites for Ste12 in a nucleosome-depleted region. In the promoters tested in this study, pFUS1 and pBAR1 do not seem to follow this rule. However, pBAR1 displays a high level of basal expression; therefore, the identified nucleosome on this promoter must be loosely bound probably allowing Ste12 binding to its target site. In pFUS1, the second strong

binding site is at the border of the nucleosome identified in a genome-wide study. More detailed measurement should be performed to assess whether Ste12 can access this site under basal conditions.

Assuming that a Ste12 dimer can be formed on the early promoters in basal condition, activation occurs rapidly via the inhibition of Dig1/2 in a manner that is proportional to the pheromone concentration and signaling activity present in the cell. Late genes do not have the ability to form these Ste12 dimers under basal conditions, because at most one consensus Ste12 site is found in a nucleosome-depleted region. Based on the evidences provided here, we postulate that the formation of a Ste12 dimer using non-consensus sites can be stabilized by Kar4. Interestingly, Kar4 has been found associated with the *AGA1* promoter in basal condition, but its deletion does not alter the level of expression or the dynamics of induction of this early promoter. However, the dynamics of induction of intermediate promoters are perturbed in a *kar4Δ* background (Fig EV5). Therefore, our data demonstrate a more global effect of Kar4 on mating genes induction than previously thought. We also observed an interaction between Ste12 and Kar4 that is strongly enhanced by pheromone treatment (Fig 3F). The association between Ste12 and Kar4 is needed to recruit Kar4 on the promoter, as in *ste12Δ* cells, Kar4 is not detected on p*AGA1* or p*FIG1* (Appendix Fig S15E and F). Kar4 presence could stabilize the TFs complex on the promoter allowing a recruitment of the chromatin remodelers, so as to evict the nucleosomes and induce an efficient transcription of the downstream ORF. The delay observed in the late gene expression is thus a combination of the requirement for Kar4 to be transcribed at sufficient levels to allow interaction with Ste12 and slow chromatin remodeling on these loci. Both Ste12–Kar4 interaction and chromatin remodeling are enhanced by MAPK activity (de Nadal & Posas, 2010), which can explain the requirement for a high pheromone concentration, and thus an elevated kinase activity, to induce the late promoters.

### Promoter induction during mating

The characterization of the various mating-dependent promoters has been performed in well-controlled conditions using synthetic mating pheromone. We next wanted to verify whether similar dynamics of gene expression occurred under the physiological conditions of mating. *MATa* cells bearing the p*FIG1* and the p*AGA1* dPSTRs were mixed on an agar pad with *MATα* cells constitutively expressing an infrared FP (tdiRFP) (Fig 4A). Strikingly, under these conditions, we also observed a clear difference in the activation of the two reporters. The *AGA1* promoter is already induced in some cells at the onset of the time-lapse (~30 min after the mixing of the mating partners). As time goes by, more cells induce the *AGA1* reporter (Fig 4A). In comparison, the p*FIG1*-dPSTR$^R$ is expressed in fewer cells and its induction precedes the fusion of the partners. Using an automated image analysis pipeline, fusion events can be detected in *MATa* cells by a strong and sudden increase in tdiRFP fluorescence (Appendix Fig S17). The single cell traces of 455 of these events recorded in one experiment were aligned temporally to their fusion time, set to 0. These quantifications reveal very clearly that the induction of p*AGA1* gradually increases until it reaches a peak prior to fusion (Fig 4B). In comparison, the *FIG1* promoter is not active until roughly 30 min before fusion. The measurements of

the response time relative to fusion confirm the kinetic difference between p*AGA1* and p*FIG1*. In addition, these new findings indicate that p*FIG1* induction seems to be tightly correlated with the fusion time, while p*AGA1* is expressed earlier and with a larger variability (Fig 4C). Cells that did not undergo fusion are highly likely to induce p*AGA1*, while p*FIG1* induction is rare in this sub-population. It can be sometimes observed in cells in the close vicinity of a set of engaged mating partners (Appendix Fig S18).

We verified that this difference in dynamics of expression is also present for other promoters (Fig 4D and E, Appendix Fig S19). In agreement with our classification based on exogenous stimulations experiments, early genes are the first ones to be induced in the mating process, followed closely by intermediate genes. Late genes induction precedes the fusion time by only 30 min, a time when cells seem committed to this process. Therefore, these genes are rarely being expressed in non-fusing cells, which is not the case for early and intermediate genes.

## Discussion

These experiments provide a better understanding of the key steps in the mating process. As soon as mating pairs are in proximity, the low level of pheromone constantly produced by the cells is sufficient to trigger a low activation of the mating pathway and induction of the expression of early mating genes. Many of these early genes are implicated in sensing and cell-fate determination and will contribute to the commitment of the partners to the mating process. If both partners are able to arrest in G1, they will extend a mating projection toward each other and polarize their sensing and secretory machinery. This will lead to a local increase in pheromone concentration that will be associated with an increase in signal transduction (Appendix Fig S20; Conlon *et al*, 2016). Mating experiments performed with a mutant unable to degrade pheromone (*bar1Δ*) clearly demonstrate that p*FIG1* induction is triggered by the concentration of pheromone sensed by the cells and not by cell–cell contacts. In this mutant, non-fusing cells activate this promoter because they experience a high concentration of pheromone independently of their proximity to a mating partner (Appendix Fig S21). The exogenous stimulation experiments have demonstrated that a step increase in pheromone concentration leads to a delayed expression of the late genes. However, in the mating process, the tight synchronization between the increase in MAPK activity, late gene expression, and fusion suggests that the early genes expressed during the sensing phase allow for a precise induction of the late genes when cells detect a further increase in mating pheromone. Taken together, our results demonstrate that yeast cells use a temporal gradient of pheromone to orchestrate the timing of expression of mating genes.

This behavior bears many similarities with morphogen sensing in development. Concentration of the diffusive signal was thought to be the key element for cell-fate decision. It is now apparent that both level and timing of morphogen stimulus dictate early and late gene expression (Gurdon *et al*, 1995, 1999; Stamataki *et al*, 2005; Dessaud *et al*, 2007; Harvey & Smith, 2009). A key question is how this temporal information is encoded to deliver the proper gene expression profile. In the simple settings offered by budding yeast, our data show that both the affinity of the TF binding sites and chromatin state at the promoter determine the concentration threshold

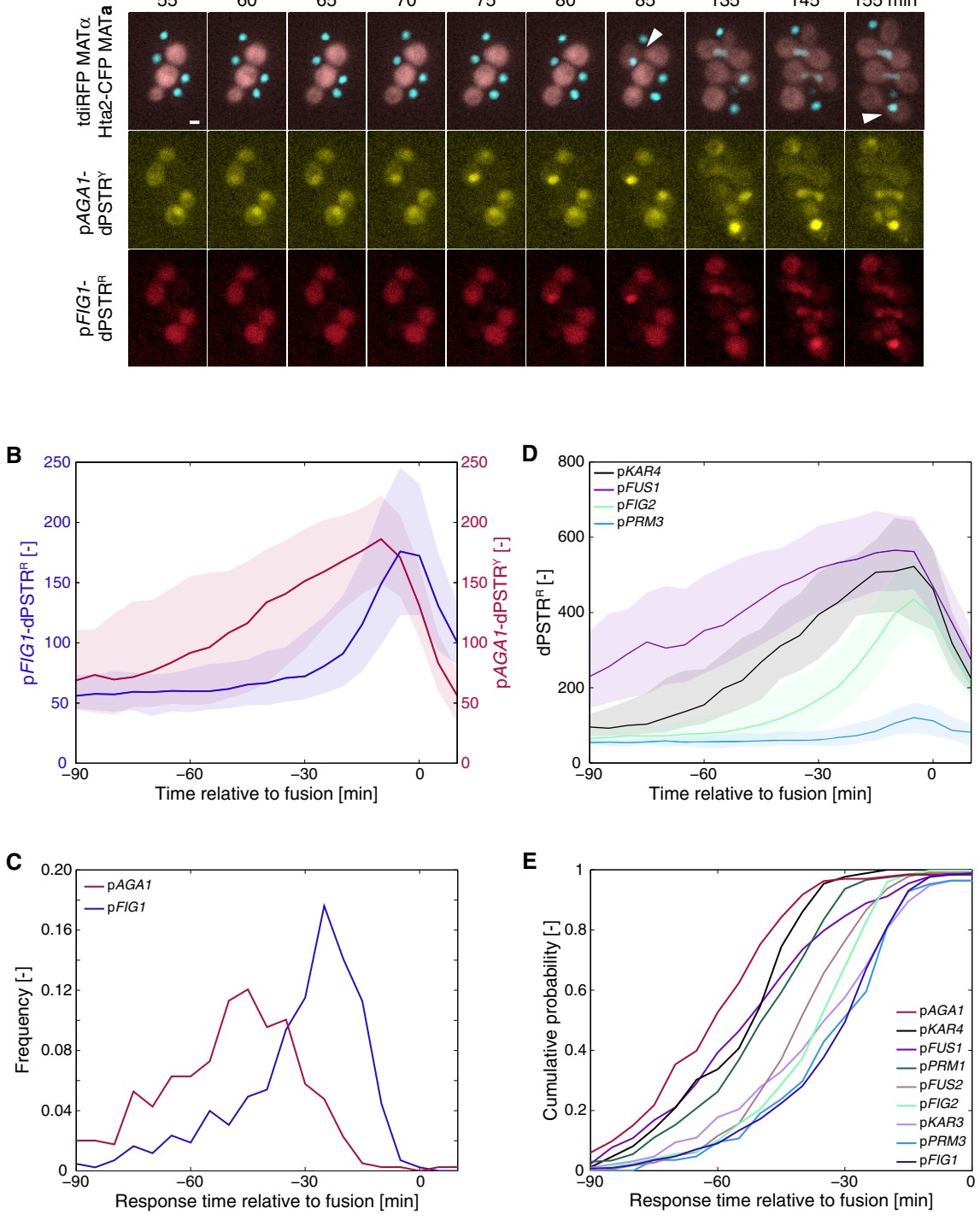

**Figure 4. Dynamics of gene expression during the mating process.**

A   Microscopy images of a mating mixture containing the MATa strain (Hta2-CFP, pFIG1-dPSTR[R], and pAGA1-dPSTR[Y]) and a MATα (cytoplasmic tdiRFP) at different times after beginning of the imaging (time 0). Fusion events are marked by a white arrow. Scale bars represent 2.5 µm.

B   Quantification of the nuclear enrichment of pFIG1-dPSTR[R] (blue, left axis) and of pAGA1-dPSTR[Y] (red, right axis). Single-cell traces were synchronized relative to their fusion time, identified by a sudden increase in tdiRFP signal into the MATa cells.

C   Distribution of the response time of pAGA1 and pFIG1 relative to the fusion time.

D   Activation dynamics of various promoters prior to fusion as measured by dPSTR[R] in different mating mixtures.

E   Cumulative probability of the response time relative to fusion for nine mating-induced promoters measured in mating conditions.

Data information: In (B and D), the solid line is the median and the shaded area represents the 25[th]–75[th] percentiles of the population.

and the timing of gene expression. This may be a general mechanism of how the timing of gene induction is orchestrated in a wide variety of cell-fate decision systems.

# Materials and Methods

### Strains and plasmids

Yeast strains and plasmids used in this study are listed in Appendix Tables S1 and S2. The dPSTR plasmids were transformed in a yeast strain from a W303 MATa background, bearing a Hta2-CFP marker (ySP580).

Each dPSTR is fully carried by a single integration vector (pSIV Wosika *et al*, 2016) and integrated in the genome. The red (and yellow) variants of the dPSTR (dPSTR$^R$ and dPSTR$^Y$, respectively) are integrated in the *URA3* (resp. *LEU2*) locus and based on interaction of the SynZips SZ1–SZ2 (resp. SZ3–SZ4) (Thompson *et al*, 2012), and the mCherry (resp. mCitrine) fluorescent variant (Aymoz *et al*, 2016). The relevant promoters of interest (typically −1,000 to −1) were amplified and cloned upstream of the inducible stable part of the dPSTR, in pSP360 for the dPSTR$^R$, and pSP363 for dPSTR$^Y$, and checked by sequencing. The inducible part was then further cloned in the pSIV vector containing the FP part of the dPSTR (pDA157 for the dPSTR$^R$ and pDA223 for dPSTR$^Y$).

For the synthetic promoter variants (Fig 3 and Appendix Fig S16), a synthetic version of p*FIG1* or p*AGA1* was designed, containing unique restriction sites (ApaI and ClaI) surrounding the region containing the PREs. This allowed to obtain dSPTR plasmids of mutants of each promoter in only one cloning (Appendix Table S2). Modified fragments of p*FIG1* and p*AGA1* with sequential mutations of PREs into NdeI (CATATG) or SnaBI (TACGTA) restriction sites were designed and synthesized by IDT (gBlocks), and cloned into pDA283 or pDA282 using *Apa*I–*Cla*I. All constructs were verified by digestion and sequencing. The integrated promoter variant was amplified from genomic DNA and sequenced for confirmation. We also verified that the presence of the cloning sites *Apa*I and *Cla*I was not altering the induction of the two promoters (data not shown).

To quantify the kinase activity, SKARS plasmids were transformed in a strain carrying p*FIG1*- or p*AGA1*-dPSTR$^R$ (Durandau *et al*, 2015).

For each transformation, eight clones were screened based on their fluorescence intensities and four clones with similar fluorescence levels were further analyzed by a time-lapse experiment upon stimulation with 1 μM of α-factor, to discard clones that would display an aberrant relocation behavior.

### Sample preparation

The cells were grown overnight in selective synthetic medium to saturation (YNB:CYN3801, CSM:DCS0031, ForMedium). They were diluted to an OD$_{600}$ of 0.05 in the morning and grown for 4 h before starting the experiment. All the time-lapse experiments were performed in well slides, for which selected wells of 96-well plates (MGB096-1-2LG, Matrical Bioscience) were coated with filtered solution of concanavalin A in H$_2$O (0.5 mg/ml, C2010-250MG, Sigma-Aldrich) for 30 min, rinsed with H$_2$O, and dried for at least

10 h. Before the experiments, the cells were diluted to an OD$_{600}$ of 0.04 and briefly sonicated, and 200 μl of cell suspension was added to a well. Imaging was started 30 min later, so as to let the cells settle to the bottom to the well. To stimulate the cells, 100 μl of a 3 μM solution of synthetic exogenous α-factor (gift from M. Peter's laboratory) was added in the well to reach a final 1 μM concentration of pheromone.

### Microscopy

Images were acquired on a fully automated inverted epi-fluorescence microscope (Ti-Eclipse, Nikon) controlled by Micro-Manager (Edelstein *et al*, 2010) and placed in an incubation chamber set at 30°C, with a 40X oil objective and appropriate excitation and emission filters. The excitation was provided by a solid-state light source (SpectraX, Lumencor). The images were recorded with a sCMOS camera (Flash4.0, Hamamatsu). A motorized XY-stage allowed recording multiple fields of view at every time point, typically five positions per well and eight wells per experiment. CFP (50 ms), RFP (300 ms), and YFP (300 ms) and two bright-field (10 ms) images were recorded at time intervals of 2 min before induction and 5 min after.

### Data analysis

Time-lapse movies were analyzed with the YeastQuant platform (Pelet *et al*, 2012). Briefly, the nuclei of the cells were segmented by thresholding of the CFP images. The contour of the cell around each nucleus was detected using two bright-field images. The cytoplasm object was obtained by removing the nucleus object expanded by two pixels from the cell object. Dedicated scripts in Matlab (The Mathworks) were written to further analyze the data. Only cells tracked from the beginning to the end of the movie were taken into consideration. In addition, a quality control was applied on each trace and only cells with low variability in nuclear and cell area, nuclear CFP fluorescence, and a ratio of RFP to YFP fluorescence lower than a certain threshold were kept for further analysis. At least 100 cells, but often 200–300 cells, were quantified for each replicate. The curves displayed in the figures are from one representative experiment out of at least three biological replicates. Appendix Table S3 summarizes the number of cells quantified in each figure panel. The nuclear enrichment values for all single-cell traces used to generate the main figures are provided in Dataset EV1.

For each cell, the difference between its average intensity in the nucleus and the cytoplasm was calculated at each time point to plot the nuclear enrichment of dPSTR$^R$ and dPSTR$^Y$.

For further analysis, all retained cell traces were smoothed by a moving average of three points. The basal level was calculated as the mean of the three time points preceding the stimulation. The corrected nuclear enrichment of the dPSTR was calculated by subtracting the basal level to the smoothed trace. The expression output represents the maximal corrected nuclear enrichment of the dPSTR. The population-averaged expression output was calculated on the mean trace of all cells. A threshold to qualify cells as expressing was defined as 20% of the population-averaged expression output. For all expressing cells, dPSTRs traces were normalized between 0 and 1, and the response time was identified as the first

time point, after stimulation, to exceed 0.2. For plots of population average correlation, and instant correlations, all cell traces were normalized by the mean trace of all cells.

The correlative promoter variability (CPV) was calculated based on the formula from Elowitz *et al* (2002) for noises as the ratio of intrinsic and total noise:

$$CPV = \frac{\eta_{int}^2}{\eta_{tot}^2} \quad \eta_{int}^2 = \frac{\langle (r_i - y_i)^2 \rangle}{2\langle r_i \rangle \langle y_i \rangle} \quad \eta_{tot}^2 = \frac{\langle r_i^2 + y_i^2 \rangle - 2\langle r_i \rangle \langle y_i \rangle}{2\langle r_i \rangle \langle y_i \rangle}$$

$r_i$ and $y_i$ are the normalized nuclear accumulations from the $i^{th}$ cell at a specific time point in the red and yellow channels, respectively. The normalization factors were obtained from the highest and lowest population-averaged intensities from the entire dataset for one replicate.

### ChIP assays

Yeast cultures were grown to early log phase ($A_{660}$ 0.4–0.6), and then, samples (50 ml) were subjected to 1 μM α-factor for the indicated times. For cross-linking, yeast cells were treated with 1% formaldehyde for 20 min at room temperature. Glycine was added to a final concentration of 330 mM for 15 min. Cells were collected, washed four times with cold TBS (20 mM Tris–HCl, pH 7.5, 150 mM NaCl), and kept at −20°C for further processing. Cell pellets were resuspended in 0.3 ml cold lysis buffer (50 mM HEPES–KOH, pH 7.5, 140 mM NaCl, 1 mM EDTA, 0.1% sodium deoxycholate, 1% Triton X-100, 1 mM PMSF, 2 mM benzamidine, 2 μg/ml leupeptin, 2 μg/ml pepstatin, 2 μg/ml aprotinin). An equal volume of glass beads was added, and cells were disrupted by vortexing (with Vortex Genie) for 13 min on ice. Glass beads were discarded, and the cross-linked chromatin was sonicated with water bath sonicator (Bioruptor) to yield an average DNA fragment size of 350 bp (range, 100–850 bp). Finally, the samples were clarified by centrifugation at 16,000 *g* for 5 min at 4°C. Supernatants were incubated with 50 μl anti-HA 12CA5 or anti-Myc 9E10 monoclonal antibodies pre-coupled to pan mouse IgG DynabeadsTM (Invitrogen, 11042). After 120 min at 4°C on a rotator, beads were washed twice for 4 min in 1 ml lysis buffer, twice in 1 ml lysis buffer with 500 mM NaCl, twice in 1 ml washing buffer (10 mM Tris–HCl pH 8.0, 0.25 M LiCl, 1 mM EDTA, 0.5% NP-40, 0.5% sodium deoxycholate), and once in 1 ml TE (10 mM Tris–HCl pH 8.0, 1 mM EDTA). Immunoprecipitated material was eluted twice from the beads by heating for 10 min at 65°C in 50 μl elution buffer (25 mM Tris–HCl pH 7.5, 1 mM EDTA, 0.5% SDS). To reverse cross-linking, samples were adjusted to 0.3 ml with elution buffer and incubated overnight at 65°C. Proteins were digested by adding 0.5 mg/ml Proteinase K (Novagen, 71049) for 1.5 h at 37°C. DNA was extracted with phenol–chloroform–isoamyl alcohol (25:24:1) and chloroform. It was finally precipitated with 48% (v/v) of isopropanol and 90 mM NaCl for 2 h at −20 °C in the presence of 20 μg glycogen and resuspended in 30 μl of TE buffer. Quantitative PCR analysis of *AGA1* and *FIG1* promoter sequences used the following primers with locations indicated by the distance from the respective ATG initiation codon: *AGA1* promoter (−310/−207); *FIG1* promoter (−400/−197); and *TEL* (telomeric region on the right arm of chromosome VI). Experiments were done on three independent chromatin preparations, and quantitative PCR analysis was done in real

time using an Applied Biosystems 7700 sequence detector. Immunoprecipitation efficiency was calculated in triplicate by normalizing the amount of PCR product in the immunoprecipitated sample by that in *TEL* sequence control. The binding data are presented as fold induction with respect to the non-treated condition.

### *In vivo* coprecipitation assay

Ste12-Myc- and/or Kar4-HA-tagged cells in mid-log phase (50 ml) were treated with 1 μM α-factor for 30 min or left untreated and then collected by brief centrifugation at 4°C. Pellets were harvested with glass beads in the FastPrep-24 (Qbiogene, 60s at speed 5) in lysis buffer A (50 mM Tris–HCl pH 7.5, 150 mM NaCl, 15 mM EDTA, 15 mM EGTA, 2 mM DTT, 0.1% Triton X-100, 1 mM PMSF, 1 mM benzamidine, 2 μg/ml leupeptin, 2 μg/ml pepstatin, 25 mM β-glycerophosphate, 1 mM sodium pyrophosphate, 10 mM sodium fluoride, 100 μM sodium orthovanadate), and lysates were clarified by centrifugation and quantified by the Bradford assay (Bio-Rad Laboratories). 1.5 mg of cleared supernatant was subjected to immunoprecipitation with rabbit polyclonal HA tag antibody (Abcam, ab9110) overnight at 4°C. Immunocomplexes were recovered with Dynabeads™ protein A (Invitrogen, 10002D) and washed with lysis buffer. Finally, they were resolved by SDS–PAGE and blotted with mouse monoclonal anti-HA 12CA5 or anti-Myc 9E10 antibodies. As a control, 50 μg of whole-cell extract was also blotted to check the expression levels of the tagged proteins (total).

### Northern blot analysis

Yeast strains were grown to mid-log phase in rich medium and then treated with 1 μM α-factor for the length of time indicated. Total RNA and expression of specific genes were probed using radiolabeled PCR fragments containing a fragment of *AGA1* ORF (+145/+936 bp), *FIG1* ORF (+106/+948 bp), and *ENO1* ORF (+1/+1,310 bp). Signals were acquired with a Fujifilm BAS-5000 PhosphorImager and ImageQuantTL software.

### MNase nucleosome mapping

Yeast spheroplast preparation and micrococcal nuclease digestions were performed as described previously with modifications (Nadal-Ribelles *et al*, 2014, 2015). Ste12-Myc and Kar4-HA double-tagged strain was grown to early log phase (A660 0.4–0.6), and samples of 500 ml of culture were exposed to 1 μM α-factor for the indicated length of time. The cells were cross-linked with 1% formaldehyde for 15 min at 30°C, and the reaction was stopped with 125 mM glycine for min. Cells were washed and resupended in 1 M sorbitol TE buffer before cell wall digestion with 100 T zymoliase (USB). Cells were then lysed and immediately digested with 60–240 mU/μl of micrococcal nuclease (Worthington Biochemical Corporation, Lakewood; NJ, USA). DNA was subjected to electrophoresis in a 1.5% (w/v) agarose gel, and the band corresponding to the mononucleosome was cut and purified using a QIAquick gel extraction kit (Qiagen). DNA was used in a real-time PCR with specific tiled oligonucleotides covering *AGA1* promoter and partial coding sequence (−928/+470) or *FIG1* promoter and partial coding sequence (−933/+463) included in Appendix Table S4. PCR quantification was referred to an internal loading control (telomeric

region in chromosome 6), and nucleosome occupancy was normalized to 1 at the ($-1$) nucleosome region of the untreated condition.

### Mating experiments

Mating experiments were performed in agar pads loaded into 96-well plates to monitor multiple strains in parallel. Agarose 2% in synthetic medium was heated 5 min at 95°C. Approximately 150 μl of this solution was placed in a small aluminum frame to form a square pad of the proper dimension to fit in a well. After cooling at 4°C for 5 min, the pad was removed from the frame. In parallel, 500 μl of cells at $OD_{600}$ 0.1 were centrifuged. MATa cells were resuspended in 10 μl of synthetic media, and this cell suspension was used to resuspend the MATα cells. 1 μl of this mixture was deposited on the agar pad and placed upside down in a well. Imaging was started roughly 30 min later, after selecting appropriate fields of view. CFP (50 ms), RFP (100 ms), YFP (100 ms), tdiRFP (100 ms), and two bright-field (30 ms) images were recorded every 5 min for 2–3 h. Cells were segmented based on the CFP and bright-field images. After quality control, cells tracked for at least 10 frames were taken into consideration for analysis. Fusion events were defined by a sudden increase of more than 50 in the average nuclear fluorescence in the tdiRFP channel. Cells were considered as non-fusing if their average nuclear fluorescence in the tdiRFP channel did not increase by more than 10 throughout the track.

### Data availability

The raw images of the time-lapse movies from the following datasets are available on the IDR repository: main figures experiments, the dose response experiments (Appendix Fig S9), and gene deletion experiments (Fig EV3, Appendix Figs S13 and S14). The DOI is 10.17867/10000114.

**Expanded View** for this article is available online.

### Acknowledgements

We thank all members of the Pelet and Martin laboratories for helpful discussions and suggestions. We thank Sophie Martin and Laura Merlini for critical readings of the manuscript. We are grateful to Clémence Varidel, Réjane Seiler, Joan Jordan, and Christine Boaron for technical assistance. Work in the Pelet laboratory is supported by the Swiss National Science Foundation (PP00P3_139121), SystemsX.ch, and the University of Lausanne. Work in the Posas and de Nadal laboratories is funded by grants from the Spanish Ministry of Economy and Competitiveness (BFU2015-64437-P and FEDER to FP; BFU2017-85152-P and FEDER to EN), the Catalan Government (2017 SGR 799), the Fundación Botín, and the Banco Santander through its Santander Universities Global Division to FP. FP is recipient of an ICREA Acadèmia (Generalitat de Catalunya).

### Author contributions

SP and DA conceived the study. SP, DA, FP, and EN designed the experiments. MS and J-JP built plasmids and strains. DA, SP, and J-JP performed the microscopy experiments and analyzed the data. CS performed biochemistry experiments. SP and DA wrote the paper. All authors read and approved the final manuscript.

### Conflict of interest

The authors declare that they have no conflict of interest.

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
