## [Review Process File · Molecular Systems Biology]

Timing of gene expression in a cell-fate decision system

Delphine Aymoz, Carme Solé, Jean-Jerrold Pierre, Marta Schmitt, Eulàlia de Nadal,
Francesc Posas & Serge Pelet

Review timeline:

Submission date:	3 October 2017
Editorial Decision:	13 November 2017
Revision received:	13 December 2017
Editorial Decision:	15 January 2018
Revision received:	15 February 2018
Accepted:	23 February 2018

Editor: Thomas Lemberger

Transaction Report:

1st Editorial Decision

13 November 2017

Thank you again for submitting your work to Molecular Systems Biology. We have now heard back from the three referees who accepted to evaluate the study. As you will see, the referees find the topic of your study of potential interest and are supportive. They raise however a series of concerns and make constructive suggestions, which we would ask you to carefully address in a revision of the present work. The recommendations provided by the reviewers are very clear in this regard.

With regard to the single cell measurements generated in this study, we would kindly ask you to make the data available, either by depositing the dataset(s) to one of the 'general' repositories such as Zenodo, Dryad or BioStudies, or by including them as Datasets to the manuscript, if size permits. If the data are deposited in a database, please list them in a Data Availability section, after Materials & Methods, that follows the example below:

```
,
#Data and software availability
```

The datasets and computer code produced in this study are available in the following databases:

- RNA-Seq data: Gene Expression Omnibus GSE46843
[<https://www.ncbi.nlm.nih.gov/geo/query/acc.cgi?acc=GSE46843>]
- Chip-Seq data: Gene Expression Omnibus GSE46748
[<https://www.ncbi.nlm.nih.gov/geo/query/acc.cgi?acc=GSE46748>]
- Protein interaction AP-MS data: PRIDE PXD000208
[<http://www.ebi.ac.uk/pride/archive/projects/PXD000208>]
- Imaging dataset: Image Data Resource doi:10.17867/10000101
[<http://doi.org/10.17867/10000101>]
- Modeling computer scripts: GitHub

[<https://github.com/SysBioChalmers/GECKO/releases/tag/v1.0>]

- [data type]: [full name of the resource] [accession number/identifier] ([doi or URL or identifiers.org/DATABASE:ACCESSION])

- Please remove the Materials & Methods from the Appendix and include them in the main text.

- As you may have noticed, we recently replaced Supplementary Information by Expanded View (EV, see examples in <http://msb.embopress.org/content/11/6/812>). In this format, a limited number of Supplementary Figures (max 5) can be integrated in the article as EV figures that are interactively collapsible/expandable and will be typeset by the publisher. In this case, the figures should be cited as 'Figure EV1, Figure EV2' etc... in the text and their respective legends should be added to the main text after the legends of regular figures. The illustrations should be provided as separate files.

- For the figures that you do NOT wish to display as Expanded View figures items, they should be bundled together with their legends in a 'traditional' supplementary PDF, now called the *Appendix*. Appendix should start with a short Table of Content and the figures should be named and referred to in the main text as: "Appendix Figure S1, Appendix Figure S2" etc. See detailed instructions regarding expanded view here: <http://msb.embopress.org/authorguide#expandedview>.

- Additional Tables/Datasets should be labeled and referred to as Table (or Dataset) EV1 etc. Table/Dataset legends can be provided in a separate tab in case of .xls files. Alternatively, you can upload a .zip file containing the Table/Dataset file and a separate README .txt file with the legend/description.

- We would also encourage you to include the source data for figure panels that show essential data, so that readers can download these data directly from the figure. Source data files must be associated to individual panels of main figures. *Numerical data* should be provided as individual .xls files (including a tab describing the data) or csv or tab-delimited text files. *For 'blots' or microscopy*, uncropped images should be submitted. For *network visualization*, Cytoscape session files, if available, can be supplied. The files should be labeled as "Source Data for Figure 1A" etc. Source Data for Expanded View and Appendix figures should be uploaded as a single ZIP file containing all the Source Data for Expanded View and Appendix content. (Additional information on source data is available in the "Guide for Authors" section at <http://msb.embopress.org/authorguide#sourcedata>).

Please resubmit your revised manuscript online, with a covering letter listing amendments and responses to each point raised by the referees. Please resubmit the paper ****within one month**** and ideally as soon as possible. If we do not receive the revised manuscript within this time period, the file might be closed and any subsequent resubmission would be treated as a new manuscript. Please use the Manuscript Number (above) in all correspondence.

When you resubmit your manuscript, please download our CHECKLIST (<http://msb.embopress.org/sites/default/files/additional-assets/EMBO%20Press%20Author%20Checklist%20-MSB.xlsx>) and include the completed form in your submission. *Please note* that the Author Checklist will be published alongside the paper as part of the transparent process <http://msb.embopress.org/authorguide#transparentprocess>.

REVIEWER REPORTS

Reviewer #1:

Aymoz et al investigate the kinetics of transcriptional activation during the response to yeast mating pheromones. They apply advanced live cell microscopy plus fluorescent sensors based on previous work from the same group. The findings convincingly establish that different pheromone-inducible promoters are activated at different times, falling into three main groups (early, intermediate, late), and that these timings are distinct from expression magnitude. Promoters in each group are activated together within individual cells, suggesting they comprise coherent but distinct transcription

programs, which is further supported by their distinct reaction to genetic perturbations. Admirably, the microscopy data are confirmed with independent assays (Northern and single-cell mRNA detection), and then are further complemented with biochemical experiments (ChIP and MNase protection) to show that early and late promoters differ in the timing of transcription factor binding and chromatin rearrangements. Finally, the authors show that these temporal transcription patterns also occur in cells undergoing dynamic communication and eventual fusion with a partner cell.

In general, this is a really lovely and impressive study. It is clear that the experiments were performed with great care and precision, and the work is substantial and thorough (including 26 dense supplemental figures). With very few exceptions, the results and interpretations are convincing, and the overall scholarship is excellent. It makes a clear advance for the field by providing compelling evidence for successive waves of transcription in this cell fate pathway, with the last set delayed until shortly before the final, irreversible event of cell-cell fusion.

The only aspect that was not fully convincing was the claim that the transcription timing can be explained by the number of consensus Ste12 binding sites in nucleosome depleted regions. See point #1 below. There are also numerous points needing clarification, listed below, but these are relatively minor. Overall, I strongly recommend publication.

Major points --

1. I was not convinced that the timing of transcription is dictated by the number of consensus Ste12 sites in nucleosome free regions. For reasons detailed below, I think the authors overstate their case (page 6). The interpretation is based on testing a limited number of variant promoters, which do not include other variants that might address alternative explanations. It seems that a clear understanding of the kinetic determinants would require a much more detailed interrogation, and probing more than one member of each class to ensure that apparent rules can be generalized from one promoter to others. In the absence of this, I suggest that the authors should be more cautious with their interpretations, and explicitly mention that they cannot rule out alternative possibilities in which the overall function relies equally strongly on nucleosomal as nucleosome-free sites, or relates to the total number of sites, or local sequence context of individual sites, or inter-site spacing, etc.
 - (a.) For example, in Supp Fig 20A-C, only the two sites in the nucleosome free region of pAGA1 were tested by mutation. If similar phenotypes were conferred by mutation of the left-most nucleosomal site as for either of the nucleosome-free sites, it would argue against the authors' model. Similarly, in Supp Fig 20D-E, if pFIG1 was affected to the same extent by mutating either of the consensus sites in nucleosomal regions (far-left and far-right), as by mutations in the nucleosome-free region, it would not support their model. (On the other hand, their model might be strengthened if such experiments suggested that the nucleosome-free sites were the key determinants.)
 - (b.) Also, when attempting to "accelerate" the FIG1 promoter, the "swap" of 150 bp from pAGA1 had a clear effect (maybe the most impressive), yet there is no obvious molecular explanation; it does not seem related to Ste12 binding sites.
 - (c.) Furthermore, the authors state (pg 6) that "...early genes possess at least two consensus binding sites for Ste12 in a nucleosome-depleted region, as assumption true for all fast promoters tested in this study except pPRM1". But from Supp Fig 15 it seems that pPRM5, pFUS1, and pBAR1 are also exceptions to this rule. So, something doesn't make sense here. It needs correction or clarification.
2. In Supp Fig 16, it seems that the dig1 mutant has a partial reduction or delay in both promoters, but this was not commented on. Was this difference found to be not statistically significant? This result is curious in light of Chou et al 2006 (Ref. 13), in which target gene expression was partly de-repressed in dig1 mutants but reduced in dig2 mutants.
3. Page 5, middle, and Fig 3D: The claim that, on the FIG1 promoter, Ste12 and Kar4 have distinct association kinetics (i.e., "Ste12 seems to accumulate first, followed by Kar4..."), is not convincing in the absence of a statistical test to verify that these minor differences are real.
4. Page 6, middle, and Supp Fig 19E-F: The text states that "in ste12Δ cells Kar4 is not detected on pAGA1 or pFIG1". Given the very large error bar (which is undefined), it seems that the effect of ste12Δ on pFIG1 would require a statistical test to argue that the means are different.

Minor points --

5. The plots of Fig 2C and Supp Fig 9B need much better explanation. These diagrams are uncommon and unintuitive, and it took me a long time to figure out what they were showing. They seem to be a sort of 3-D plot compressed into 2-D, but initially it is unclear what is displayed on each axis or what the curves represent. To clarify, I suggest providing a simple and explicit statement that the two axes show normalized expression levels (rather than time, which is the other element of the display). Then, provide a clearer statement that each line shows the progression of time, going from left to right. (The current language -"evolves towards upper right corner as promoters are being induced" - is indirect and confusing.) It would also help to reiterate these descriptions in the Legend as well as in the text.

6. Page 4, middle: References to the cell cycle are unclear.

(a.) It is unclear what is meant by the following statement: "...due to the asynchronous induction of pAGA1 and pFAR1 during the cell cycle...". Is this intended to imply that basal expression is constant for pAGA1 but periodic for pFAR1? If so, perhaps state that directly.

(b.) Two comments about the cell cycle cite Supp Fig 10 ("...pAGA1 and pFAR1 during the cell cycle..." and "...KAR3 induction during the cell cycle..."). This is confusing: how does Supp Fig 10 illustrate anything about cell cycle expression? These comments need clarification.

7. Page 4, middle: "For the two slow promoters pFIG1 and pKAR3, the basal CPV value..." and "After stimulus, this CPV value..." It is ambiguous here whether these comments are referring to comparisons of EACH promoter to pAGA1 (i.e., as in Supp Fig 9C) or to comparisons of the two promoters to each other (i.e., as in Fig 2F, green line).

8. Page 4, middle: "...while late gene expression can be delayed by more than an hour." It is unclear how "more than an hour" is derived. Fig 2B and Supp Fig 9A suggest the late genes are delayed by roughly 20-30 min compared to pAGA1.

9. Supp Fig 7A-B: the legend needs to define the red and green signals. (Presumably green is PP7-GFP, but what is red?)

10. The blot in Supp Fig 7E is the same as in Supp Fig 19A, but with the right half hidden. There should be a note added to one of the legends to indicate that this is a duplicate and not independent.

11. Regarding Supp Fig 10:

(a.) the inset keys say "x % Basal EC". The legend should define "EC" or cite where it is defined elsewhere.

(b.) the % values in parentheses are not defined. (I.e., % of what? % of max. induced level?)

(c.) pFUS1 is in the "30-80% basal" group; is this supposed to imply that the mean basal levels are 30-80% of the induced levels? If so, that seems surprising, based on literature and in comparison to data in Supp Fig 12A-B. Some clarification is required.

12. Regarding Supp Fig 15:

(a.) for pSTE12, why is the left-most site colored green rather than red? It seems to match the consensus defined in the legend, and is as good a match as other red sites in other promoters.

(b.) in pKAR3, two of the green arrows are dashed; what is this meant to imply?

13. The number of trials (n) and the nature of error bars (e.g., SD, SEM, etc.) need to be defined in the legend for the following Figures: Fig 3C-D; Supp Fig 11 C-D; Supp Fig 19E

14. Legends should note at what TIME measurements are taken for the following Figures: Supp Fig 11A-C; Supp Fig 12; panels C of Supp Figs 16-18.

15. Supp Fig 20G is missing the pink curve (there are only 4 curves, not 5).

16. Typographical errors:

(a.) Page 1: "activate the mating MAPKs few minutes" should say "a few minutes".

- (b.) Page 1, bottom: "within the 30 minutes" - delete "the".
- (c.) Page 5, middle: "eviction of the -1 histone" should say "-1 nucleosome".
- (d.) Supp Fig 4A legend, last line: "arbitrary" should be "arbitrarily".
- (e.) Supp Fig 9B and 9C legend: "doted line" should be "dotted".
- (f.) Supp Fig 10 legend: "the orange doted line on all plot" should say "dotted" and "plots".

Reviewer #2:

Review of "Timing of gene expression in a cell-fate decision system" by Aymoz et al.

In this manuscript, the authors use their recently developed fast reporters of MAPK activity and reporter expression to study the behavior of mating pheromone response genes. They find variable delays between MAPK activation and the initiation of gene expression of these approximately 10 genes. Genes fall into roughly three groups: early, intermediate and late genes. In trying to explain the reason for the delay, they analyze the promoter architectures, Ste12 binding sites and nucleosome positioning. They discover that late promoters do not have pairs of consensus Ste12 binding sites in nucleosome depleted regions. Instead, they have one "good site" next to a non-consensus, poorer site for Ste12, and only one site is available for binding without prior nucleosome removal. They show that part of the delay is indeed due to the time required to open the promoter for Ste12 binding. Notably, they find that one of the early response genes, Kar4, is important for inducing the late genes that lack a perfect pair of sites, and they speculate that this is because Ste12-Kar4 dimers might bind better to pair of sites with one good and one non-consensus site. Mutation of the non-consensus site to a perfect Ste12 site renders the promoter Kar4 independent. Finally, they show that the dynamics observed using synthetic pheromone is also apparent in mating mixes, where the late genes seem to come up just before fusion.

The manuscript has much more information than what I summarized above, such as rather interesting study on noise of the various promoters and their correlated expression. The experiments are of very high quality, and really insightful. I recommend publication with minor revisions.

Points to address:

- 1-Has the data been uploaded somewhere? In this paper, there is a lot of single cell data that would be very valuable for other researchers. All should be uploaded to a suitable repository.
- 2-The critical experiment shown in Fig3E does not lead to a very strong conclusion. By mutating the non-consensus site to a good consensus and simultaneously exchanging the first 150 bp of one promoter for the other, they mildly speed up promoter induction. What is the authors speculation?
- 3-The introduction does not have a proper review of previous work on the timing of gene expression in response to pheromone. There are several works where rtPCR (even northern blots) have been used in short pheromone exposures. For example, in 1994 the lab of Fred Cross published a now classic paper where they stimulated with pheromone for 12 minutes at various times after release from cell cycle arrest. In it they see strong FUS1 mRNA expression. Another example, microarray data collected in 2000 by Roberts et al should be a good place to find the dynamics of gene expression and compare with this manuscript results. And there are others.
- 4-When only the expression level of a reporter is shown (and not the whole time course), it is not clear at which time reporter expression was measured (for one important example, see FigS11).
- 5-Also it is not clear if the non responders (a large fraction for the FIG1 reporter at low stimulus) are included in the average measures shown in the paper (again for example in FigS11). The dose responses will look very different one way or the other.
- 6-The authors comment on the steepness of the dose responses. To be more accurate, it would be very useful if they provide a measure of steepness, such as hill coefficient. Again, I suppose the resulting measure will be affected if non-responders are or not included.
- 7-How do the authors explain that for AGA1 there is almost no difference between WT and bar1-delta? FIG1 shows the expected large difference.
- 8-It is a pity the authors did most of the measurements in BAR1+ cells. The effect of Bar1 is not relevant as to the sensitivity of each promoter, and due to the effect of Bar1 in late promoters, we are left unable to compare their relative sensitivity to pheromone. As far as it is believed in the field, the only action of Bar1 is to destroy pheromone. So, doing the experiments in this way complicates the interpretation of the results.

Reviewer #3:

In this manuscript, Aymoz et al. studied the dynamics of the yeast molecular response to mating pheromone, using reporter systems that they previously developed. These systems (called dPSTR) are based on the relocalization of a fluorescent reporter and they offer high temporal resolution and allow to track individual live cells over time. In the present study, they monitored the transcriptional induction of 14 mating genes after exposure to pheromone and they report different kinetics of activation, with early (eg pAGA1), middle(eg pFUS2) and later promoters (eg pFIG1) being separated by up to 30 minutes in time. They controlled that this actually corresponded to delayed transcriptional activations and not technical artefacts, and that similar delays occurred in cells undergoing physiological mating (and not just in cells responding to artificial addition of pheromone). They conducted a genetic study of these delays, by either manipulating binding sites in the promoters, or by studying the response in null mutants of putative transcriptional regulators. This led to the conclusion that KAR4 acts as a relay between the early activation of AGA1 and the late activation of FIG1. Chromatin IP and MNase-protection assays demonstrated the progressive remodelling at promoters.

This is an impressive study that illustrates how a global response is in fact structured (sequential) in time. Experiments are very precise (in time and in quantification), the authors provide data that is abundant (many cells), accompanied by important controls (Northern blots, biochemical assays) and they verified the biological relevance of the different timings (mating assays). This detailed temporal description of the mating response will be interesting for the specialized readership but also as an illustration of eukaryotic signaling responses in general. It also shows the power of the dPSTR reporters that many other labs may want to use.

Major point

The authors may have sub-exploited the wealth of data they produced with the two-colors experiments. For each cell, a time-course quantification is available for 2 promoters, but the figures seem to report only the correlation across identical time points, or across the entire data. Have the authors explored cross-correlation functions? For any pair of promoters they studied (pAGA1, pX) or (pFIG1,pKAR3), they could generate a population of such functions (one per cell). Each function may be imprecise but, together, cross-correlation functions may peak at a critical time delay. If they do, this would correlate early AGA1 signals with later promoter induction in a deterministic manner, and estimate the corresponding delay. If they do not, this would support a stochastic secondary activation of the late gene. Same is true for (Ste7-SKARS, pAGA1) and (Ste7-SKARS, pFIG1) data.

Minor points

- Would promoter swapping of the FIG1 and AGA1 gene have any effect on mating efficiency? The manuscript is already very dense and this could be a later follow-up characterization. However, if generating strains and running a competitive mating assay is do-able, this would evaluate the biological importance of a proper sequence of events.

- The text is a mixture of active mode "We quantified... We used.." and passive mode "Promoters were compared... were characterized...". The authors should choose one and stay consistent throughout the manuscript. Same comment about the use of present and past tense.

- Colors for times 0 and 60 min are hard to distinguish on plots 1G, 2D-E, S13-S14, and it would be informative to see a subset of time trajectories linking points of the same cells. For example on 1G, are the few dots FIG1+/AGA- from the same cells? Are they technical errors or are there few cells violating the general rule ?

- The differential kinetics of mRNA increase (Fig S7E or S7C-D) should appear in main Fig. 1

(space is available).

- Intro: "in embryonic development": sounds very restrictive for a first sentence. Immune response, tissue repairs... are also concerned.

- Intro: Two or three sentences reminding the SKARS and dPSTR systems could be added before paragraph "In this study...".

- p2: sentence "Analysis of ... still results in a delayed and broad..." is badly written.

- p2: signaling competent cells -> signaling-competent cells.

- p3: "the population moves upwards" has no biological meaning. Simply say there is a first shift along x and a later shift along y.

- p3: simultaneous activation of all mating genes -> simultaneous transcriptional activation of all mating genes

- p5: gnc5/SAGA is not strictly a remodeler. Change "nucleosome remodeling" by "nucleosome modification"

- p5: -1 histone -> -1 nucleosome.

- p6: "turned out to be" -> "was"

- p6: "accelerated the induction of the dPSTR" -> "accelerated the induction response"

- Writing "temporal gradient of pheromone" (p7) is imprecise and unnecessary. The data of bar1-cells show the importance of pheromone concentration, but not a temporal gradient.

- Ref 19 is incorrectly referenced (NRG and not NPG).

-Supp Fig 5: I suggest to change the figure title "The kinetics differences between promoters is not an artifact due to the dPSTR FP and SynZip pair." by "Consistent differential kinetics between promoters after inverting dPSTR pairs."

- Legend supp Fig 9: Signtest -> Sign test.

- Colors supp fig 10D are hard to distinguish in the legend: put names next to lines?

- Instead of 4 panels of supp fig 10, it may be clearer to show a dot plot with y = response time (values of x-axis of 2A) and x = basal EC in %.

General Revisions

1. We have included additional experiments performed with mutated promoters to validate the key importance of some Ste12 binding sites relative to other consensus sites in Appendix Figure S16. For instance, in pFIG1, mutation of the second consensus binding site abolishes completely the expression from this promoter variant. The exact same mutation performed on the two other consensus sites present on this promoter reduce gene expression but doesn't affect the expression dynamics.
2. We now present the gene expression phenotype of the *dig1Δ*, *dig2Δ* and the *dig1Δdig2Δ* mutants in Appendix Figure S13 and S14. We have modified the text to explain our hypothesis regarding the unexpected behavior of the *dig1Δ* mutant.
3. We have fitted a Hill curve on the dose responses performed with the various promoters tested. The results from these analyses are presented in a small table integrated to Appendix Figure S9. We obtain a Hill coefficient close to 3 for the three late promoter tested. As comparison, for early and intermediate promoters, the Hill coefficient obtained from the fit is close to 1.
4. We have clarified in the text and in the figure legends the number of measurements that we have used in this study. In addition, we have added an expended view table listing for all main figures and for each strain the number of single cell traces quantified.
5. Finally, we are in the process of uploading the raw data from this study (images and quantification) to the IDR repository.

Point-by-Point response

Reviewer #1:

Aymoz et al investigate the kinetics of transcriptional activation during the response to yeast mating pheromones. They apply advanced live cell microscopy plus fluorescent sensors based on previous work from the same group. The findings convincingly establish that different pheromone-inducible promoters are activated at different times, falling into three main groups (early, intermediate, late), and that these timings are distinct from expression magnitude. Promoters in each group are activated together within individual cells, suggesting they comprise coherent but distinct transcription programs, which is further supported by their distinct reaction to genetic perturbations. Admirably, the microscopy data are confirmed with independent assays (Northern and single-cell mRNA detection), and then are further complemented with biochemical experiments (ChIP and MNase protection) to show that early and late promoters differ in the timing of transcription factor binding and chromatin rearrangements. Finally, the authors show that these temporal transcription patterns also occur in cells undergoing dynamic communication and eventual fusion with a partner cell.

In general, this is a really lovely and impressive study. It is clear that the experiments were performed with great care and precision, and the work is substantial and thorough (including 26 dense supplemental figures). With very few exceptions, the results and interpretations are convincing, and the overall scholarship is excellent. It makes a clear advance for the field by providing compelling evidence for successive waves of transcription in this cell fate pathway, with the last set delayed until shortly before the final, irreversible event of cell-cell fusion.

The only aspect that was not fully convincing was the claim that the transcription timing can be explained by the number of consensus Ste12 binding sites in nucleosome depleted regions. See

point #1 below. There are also numerous points needing clarification, listed below, but these are relatively minor. Overall, I strongly recommend publication.

Major points --

1. I was not convinced that the timing of transcription is dictated by the number of consensus Ste12 sites in nucleosome free regions. For reasons detailed below, I think the authors overstate their case (page 6). The interpretation is based on testing a limited number of variant promoters, which do not include other variants that might address alternative explanations. It seems that a clear understanding of the kinetic determinants would require a much more detailed interrogation, and probing more than one member of each class to ensure that apparent rules can be generalized from one promoter to others. In the absence of this, I suggest that the authors should be more cautious with their interpretations, and explicitly mention that they cannot rule out alternative possibilities in which the overall function relies equally strongly on nucleosomal as nucleosome-free sites, or relates to the total number of sites, or local sequence context of individual sites, or inter-site spacing, etc.

(a.) For example, in Supp Fig 20A-C, only the two sites in the nucleosome free region of pAGA1 were tested by mutation. If similar phenotypes were conferred by mutation of the leftmost nucleosomal site as for either of the nucleosome-free sites, it would argue against the authors' model. Similarly, in Supp Fig 20D-E, if pFIG1 was affected to the same extent by mutating either of the consensus sites in nucleosomal regions (far-left and far-right), as by mutations in the nucleosome-free region, it would not support their model. (On the other hand, their model might be strengthened if such experiments suggested that the nucleosome-free sites were the key determinants.)

This is a valid point raised by the reviewer. The model was also based on experiments performed with additional promoter variants. In the first version of the manuscript, we included only the data from the variants that strongly affected the induction of the promoter. We have now added other control measurements to the new Appendix Figure S16. For pAGA1, mutation of the leftmost Ste12 binding site barely affects the dynamics and amplitude of the response, while mutation of either site in the nucleosome depleted region slows down the response. Mutating both of these sites simultaneously renders the promoter almost unresponsive.

For pFIG1, mutation of either of the two consensus Ste12 binding sites hidden by nucleosomes under basal conditions do not alter the dynamics of expression of the dPSTR but decreases the amplitude of the response. In comparison, mutating either the consensus and non-consensus site in the nucleosome depleted region abolishes the response of the promoter. We believe that these data substantiate our model and strengthen the importance of the binding sites in a nucleosome depleted region for both early and late promoters.

(b.) Also, when attempting to "accelerate" the FIG1 promoter, the "swap" of 150 bp from pAGA1 had a clear effect (maybe the most impressive), yet there is no obvious molecular explanation; it does not seem related to Ste12 binding sites.

pAGA1 is slightly induced in a cell-cycle dependent manner while pFIG1 is constantly repressed under normal growth conditions. This difference should be reflected in the nucleosome occupancy on each one of these promoters. Our hypothesis is that by using the last -150 bp of pAGA1 on pFIG1 or vice-versa, we confer some of these nucleosome properties to the chimeric promoters and thereby affect their expression dynamics. We have modified the text to explain more clearly our hypothesis.

(c.) Furthermore, the authors state (pg 6) that "...early genes possess at least two consensus binding sites for Ste12 in a nucleosome-depleted region, as assumption true for all fast promoters tested in this study except pPRM1". But from Supp Fig 15 it seems that pPRM5, pFUS1, and pBAR1 are also exceptions to this rule. So, something doesn't make sense here. It needs correction or clarification.

The reviewer is correct. Our current model is based mainly on data from pAGA1 and pFIG1 and it is difficult to extend it to other promoters. One simple rule that seems to apply to most early promoter is the presence of two Ste12 binding sites in a nucleosome depleted region. As noticed by the reviewer, one of the pBAR1 site seems protected by a nucleosome. However, since BAR1 has an important basal expression level, we consider that the nucleosome found on this promoter should be rather loose and therefore a Ste12 dimer should be able to form on this promoter in basal conditions. Regarding pFUS1, the second consensus Ste12 binding site is found at the border of the nucleosome identified in the large-scale dataset from Brogaard et al. More detailed analysis might reveal that the Ste12 site falls actually outside of the nucleosome region. Indeed, in our MNase experiment, we failed to confirm the presence of one of the nucleosomes predicted to exist on the AGA1 promoter.

We have changed the text to better explain our hypothesis and why we consider that pFUS1 and pBAR1 might still possess two Ste12 bound to the promoter under basal conditions.

2. In Supp Fig 16, it seems that the *dig1* mutant has a partial reduction or delay in both promoters, but this was not commented on. Was this difference found to be not statistically significant? This result is curious in light of Chou et al 2006 (Ref. 13), in which target gene expression was partly de-repressed in *dig1* mutants but reduced in *dig2* mutants.

For completeness, we have added the *dig2* Δ mutant in Appendix Figure S13 and moved the *dig1* Δ mutant to Appendix Figure S14. The *dig2* Δ deletion does not affect expression of pAGA1 or pFIG1 and therefore belongs to the Group I. As pointed out by the reviewer, the *dig1* Δ deletion affects both pAGA1 and pFIG1 expression in a similar manner and therefore belongs to the Group II. This difference is clearly significant.

It has been already noticed previously that the *dig2* Δ phenotype is usually minor probably because the expression level of Dig2 is lower than the one of Dig1 (Olson et al. Molecular and Cellular Biology, 2000 and Pelet Scientific Reports, 2017). The *dig1* Δ *dig2* Δ phenotype provides the expected derepression of the mating dependent promoters. Therefore, it is hard to understand the *dig1* Δ phenotype which displays a delay in induction of the promoter, thereby suggesting a stronger repression of the mating promoters in this mutant. One putative explanation for this phenotype is that the strong induction of the filamentous genes in this background decreases the pool of available Ste12 for mating genes expression. Note that the data presented in Figure 2 in Chou et al 2006 represent only the basal expression level in various mutants. The PRE-LacZ data are the ones that should be compared to our dataset and for this artificial promoter, the derepression in *dig1* Δ is really minor. To clarify this point, we have modified the text to discuss the *dig1* Δ phenotype.

3. Page 5, middle, and Fig 3D: The claim that, on the FIG1 promoter, Ste12 and Kar4 have distinct association kinetics (i.e., "Ste12 seems to accumulate first, followed by Kar4..."), is not convincing in the absence of a statistical test to verify that these minor differences are real.

The reviewer is completely right, thanks for noticing our mistake. There is no clear difference on the recruitment of Ste12 and Kar4 on pFIG1. In contrast, the association of

both factors on pFIG1 and pAGA1 are significantly different. We have now corrected the manuscript accordingly.

4. Page 6, middle, and Supp Fig 19E-F: The text states that "in *ste12Δ* cells Kar4 is not detected on pAGA1 or pFIG1". Given the very large error bar (which is undefined), it seems that the effect of *ste12Δ* on pFIG1 would require a statistical test to argue that the means are different.

The reviewer noticed the large error bar corresponding to Kar4 occupancy on pFIG1 in the wild type strain presented in Supp Fig 19E (right lower graph). The error bar in this point is a mistake of duplicating the error of the same sample on pAGA1 (2.00). We checked all the data and the means are correct and the SD corresponding to this promoter is 0.46.

The raw data is presented in the table below and the corrected value is highlighted in bold:

Strain	Basal Kar4 occupancy on pAGA1 (mean ± SD)	Basal Kar4 occupancy on pFIG1 (mean ± SD)
No tag	1 ± 0.07	1 ± 0.00
WT	22.46 ± 2.00	2.44 ± 0.46
ste12Δ	0.66 ± 0.05	1.06 ± 0.27

In addition, we now performed a Student's t test between wt and *ste12Δ* strains to demonstrate that Kar4 occupancy is statistically significant at both promoters: pAGA1 (p-value=6.056E-07) and pFIG1 (p-value= 0.00393), confirming our initial statement.

Minor points --

5. The plots of Fig 2C and Supp Fig 9B need much better explanation. These diagrams are uncommon and unintuitive, and it took me a long time to figure out what they were showing. They seem to be a sort of 3-D plot compressed into 2-D, but initially it is unclear what is displayed on each axis or what the curves represent. To clarify, I suggest providing a simple and explicit statement that the two axes show normalized expression levels (rather than time, which is the other element of the display). Then, provide a clearer statement that each line shows the progression of time, going from left to right. (The current language - "evolves towards upper right corner as promoters are being induced" - is indirect and confusing.) It would also help to reiterate these descriptions in the Legend as well as in the text.

We have modified the main text and the figure legend to clarify this point.

6. Page 4, middle: References to the cell cycle are unclear.

(a.) It is unclear what is meant by the following statement: "...due to the asynchronous induction of pAGA1 and pFAR1 during the cell cycle...". Is this intended to imply that basal expression is constant for pAGA1 but periodic for pFAR1? If so, perhaps state that directly.

AGA1 and FAR1 are both cell cycle regulated genes (Oehlen et al. Mol Cell Biol, 1996). However, they are expressed in different phases of the cell cycle. Therefore, under normal conditions, we detected an absence of correlation between their expression levels, which

resulted in a large CPV value. Upon induction, however, this heterogeneity disappeared because both promoters are expressed strongly and the CPV drops rapidly. The text has been modified to clarify this point.

(b.) Two comments about the cell cycle cite Supp Fig 10 ("...pAGA1 and pFAR1 during the cell cycle..." and "...KAR3 induction during the cell cycle..."). This is confusing: how does Supp Fig 10 illustrate anything about cell cycle expression? These comments need clarification.

It turns out that many mating genes are expressed at low levels in a cell cycle dependent manner (Oehlen et al. Mol Cell Biol, 1996). Appendix Figure S8 displays the basal induction levels of the 14 tested promoters and therefore reports for some of them on the level of this cell cycle dependent expression. As stated above, we detect a large basal level for pAGA1 and pFAR1 because of their cell cycle-dependent induction. pKAR3 also has some basal expression that is cell cycle regulated. We have modified the text to describe these phenomena.

7. Page 4, middle: "For the two slow promoters pFIG1 and pKAR3, the basal CPV value..." and "After stimulus, this CPV value..." It is ambiguous here whether these comments are referring to comparisons of EACH promoter to pAGA1 (i.e., as in Supp Fig 9C) or to comparisons of the two promoters to each other (i.e., as in Fig 2F, green line).

It refers to the comparison of the two slow promoters to each other. The text has been modified to make it clearer.

8. Page 4, middle: "...while late gene expression can be delayed by more than an hour." It is unclear how "more than an hour" is derived. Fig 2B and Supp Fig 9A suggest the late genes are delayed by roughly 20-30 min compared to pAGA1.

The reviewer is correct in saying that on average, the delay between pAGA1 and pFIG1 induction is 20-30 minutes. In the sentence mentioned in his comment, we referred to the general dynamic of late gene induction relative to the pheromone stimulus. According to Figure 1F, some cells induce pFIG1 more than an hour after the stimulus. We have modified the text to avoid any confusion.

9. Supp Fig 7A-B: the legend needs to define the red and green signals. (Presumably green is PP7-GFP, but what is red?)

In Appendix Figure S7, the green signal corresponds indeed to the PP7-2xGFP and the red fluorescence arises from a Hta2-mCherry tag present in the strain. The legend has been updated to clarify these points.

10. The blot in Supp Fig 7E is the same as in Supp Fig 19A, but with the right half-hidden. There should be a note added to one of the legends to indicate that this is a duplicate and not independent.

The legend of Appendix Figure S15 has been updated.

11. Regarding Supp Fig 10:

(a.) the inset keys say "x % Basal EC". The legend should define "EC" or cite where it is defined elsewhere.

(b.) the % values in parentheses are not defined. (I.e., % of what? % of max. induced level?)

(c.) pFUS1 is in the "30-80% basal" group; is this supposed to imply that the mean basal levels are 30-80% of the induced levels? If so, that seems surprising, based on literature and in comparison to data in Supp Fig 12A-B. Some clarification is required.

EC refers to Expressing Cells and the percentage calculated represents the fraction of cells in the population that displays an induction of the promoter under basal conditions. The threshold for detecting this basal expression has been obtained from a strain which contains only the mCherry-SZ2 and not the inducible part of the dPSTR resulting in a uniform distribution of the fluorescence throughout the cell. For pFUS1, the value of 34 given in the legend therefore means that 34% of the cells display an expression level in basal condition that overcomes the expected uniform distribution of the mCherry protein and are thus considered as expressing in absence of pheromone stimulation i.e. in basal conditions. The legend of Appendix Figure S8 has been edited to explain this more clearly.

12. Regarding Supp Fig 15:

(a.) for pSTE12, why is the leftmost site colored green rather than red? It seems to match the consensus defined in the legend, and is as good a match as other red sites in other promoters.

The reviewer is correct, we have modified the figure accordingly.

(b.) in pKAR3, two of the green arrows are dashed; what is this meant to imply?

These dashed arrows represented other putative Ste12 binding site (far from the consensus) that were reported by Kurihara et al. (Molecular and Cellular Biology, 1996). We had annotated them to see if they could provide some insights into the dynamics of pKAR3 induction. For simplification, we have now removed them.

13. The number of trials (n) and the nature of error bars (e.g., SD, SEM, etc.) need to be defined in the legend for the following Figures: Fig 3C-D; Supp Fig 11 C-D; Supp Fig 19E

The legends have been updated to describe the number of replicas of the biochemical experiments from Figure 3 C and D and Appendix Figure S15. Those experiments were performed in triplicate and that the error bars represent the standard deviations over the three experiments.

For the fraction of responding cells in Expanded View 3, we have shown the data from a single representative replicate. But error bars are provided for similar measurements in Appendix Figure S9

14. Legends should note at what TIME measurements are taken for the following Figures: Supp Fig 11A-C; Supp Fig 12; panels C of Supp Figs 16-18.

For these figures, the value plotted is the expression output. For each single cell trace, the expression output (EO), defined in Appendix Figure S4, represents the difference between the maximum of the trace and the basal value (average of first three time points before stimulation). To simplify the understanding of the manuscript, a short definition of the expression output has been repeated in all the legends of the figures where this metric is used.

15. Supp Fig 20G is missing the pink curve (there are only 4 curves, not 5).

Thank you for noticing this mistake. The curve has been added to the panel.

16. Typographical errors:

(a.) Page 1: "activate the mating MAPKs few minutes" should say "a few minutes".

(b.) Page 1, bottom: "within the 30 minutes" - delete "the".

- (c.) Page 5, middle: "eviction of the -1 histone" should say "-1 nucleosome".
(d.) Supp Fig 4A legend, last line: "arbitrary" should be "arbitrarily".
(e.) Supp Fig 9B and 9C legend: "doted line" should be "dotted".
(f.) Supp Fig 10 legend: "the orange doted line on all plot" should say "dotted" and "plots".

We have corrected the text to remove these errors. Thank you for noticing them.

Reviewer #2:

Review of "Timing of gene expression in a cell-fate decision system" by Aymoz et al.

In this manuscript, the authors use their recently developed fast reporters of MAPK activity and reporter expression to study the behavior of mating pheromone response genes. They find variable delays between MAPK activation and the initiation of gene expression of these approximately 10 genes. Genes fall into roughly three groups: early, intermediate and late genes. In trying to explain the reason for the delay, they analyze the promoter architectures, Ste12 binding sites and nucleosome positioning. They discover that late promoters do not have pairs of consensus Ste12 binding sites in nucleosome depleted regions. Instead, they have one "good site" next to a non-consensus, poorer site for Ste12, and only one site is available for binding without prior nucleosome removal. They show that part of the delay is indeed due to the time required to open the promoter for Ste12 binding. Notably, they find that one of the early response genes, Kar4, is important for inducing the late genes that lack a perfect pair of sites, and they speculate that this is because Ste12-Kar4 dimers might bind better to pair of sites with one good and one non-consensus site. Mutation of the non-consensus site to a perfect Ste12 site renders the promoter Kar4 independent. Finally, they show that the dynamics observed using synthetic pheromone is also apparent in mating mixes, where the late genes seem to come up just before fusion. The manuscript has much more information than what I summarized above, such as rather interesting study on noise of the various promoters and their correlated expression. The experiments are of very high quality, and really insightful. I recommend publication with minor revisions.

Points to address:

1-Has the data been uploaded somewhere? In this paper, there is a lot of single cell data that would be very valuable for other researchers. All should be uploaded to a suitable repository.

We will supply to the editor the single cell traces used to generate all graphs in the main figures. These data should therefore be directly available from the manuscript web page. In addition, we have contacted IDR (<http://idr.openmicroscopy.org/>) and they are interested to host the raw images and the quantifications from our experiments. The documentation and the transfer of these data will however require some time.

2-The critical experiment shown in Fig3E does not lead to a very strong conclusion. By mutating the non-consensus site to a good consensus and simultaneously exchanging the first 150 bp of one promoter for the other, they mildly speed up promoter induction. What is the authors speculation?

The DNA sequence influences the affinity of the nucleosome for a specific region. Because *AGA1* is expressed in control conditions, we reasoned that the nucleosomes may be more labile on this promoter compared to the *pFIG1* promoter which is constantly repressed under normal growth condition. Thereby, by transferring to *pFIG1* the 150 bp of *pAGA1* which include the region where the -1 nucleosome binds, we believed that we facilitated the remodeling of the chromatin on *pFIG1* and in doing so accelerated the gene expression. We have improved the text to explain our hypothesis better.

3-The introduction does not have a proper review of previous work on the timing of gene expression in response to pheromone. There are several works where *rtPCR* (even northern blots) have been used in short pheromone exposures. For example, in 1994 the lab of Fred Cross published a now classic paper where they stimulated with pheromone for 12 minutes at various times after release from cell cycle arrest. In it they see strong *FUS1* mRNA expression. Another example, microarray data collected in 2000 by Roberts et al should be a good place to find the dynamics of gene expression and compare with this manuscript results. And there are others.

We don't think that the work mentioned on *FUS1* provides much insight into the dynamics of gene expression from mating dependent promoters. It only proves that mRNA is present already 12 minutes after stimulus but it does not say anything about the evolution of expression as function of time or how other promoters are expressed.

The work from Roberts et al. provides many insights into the expression program induced by pheromone and how it is affected by various deletions and we cited it multiple times in our manuscript. However, in terms of dynamic of gene expression, the information provided by this study is very limited. Rebuttal Figure 1 displays the dynamics measured for three early and three late genes in the Roberts et al dataset. You can observe that the dynamics are completely different. *FIG1* for instance displays already more than 60% induction after 15 minutes. This difference is probably due to the fact that microarray measurements are based on fold induction compared to the uninduced state. Since *FIG1* is repressed under basal condition, a very slight increase in mRNA level results in a high fold induction.

Rebuttal Figure 1. Raw data extracted from Roberts et al. displaying the dynamics of mRNA induction of three early (solid lines) and 3 late genes (dashed line).

4-When only the expression level of a reporter is shown (and not the whole time course), it is not clear at which time reporter expression was measured (for one important example, see FigS11).

We have called this metric expression output (EO) and it corresponds to the difference between the maximum intensity reached by the single cell traces and the basal level (mean of three points before stimulus). This measurement is presented in the Appendix Figure S4. We have also added this information to the legends of each figure where this metric is used.

5-Also it is not clear if the-non responders (a large fraction for the *FIG1* reporter at low stimulus) are included in the average measures shown in the paper (again for example in FigS11). The dose responses will look very different one way or the other.

In general, all the single cell traces were considered and displayed in the figures. The two exceptions are the following:

- Response time (for instance: Figure 1F or Figure 2 A): In order to define a response time, the cell has to be expressing the promoter of interest.

- Difference of Response time (for instance: inset Figure 1F or Figure 2B): In this case only the sub-population of cells expressing both promoters can be used for proper comparison

We have clarified this point in the legend of the figures where we used these measurements. We also added an expanded view Table 3 containing the number of cells from each experiment and the number used in each plot, which reflects the proportion of cells expressing the promoters.

Regarding the dose-response experiments presented in Expanded View 3 or Appendix Figure S9, all cells were used. We have generated the curves of the dose response with only the responding cells. The general picture does not change much, because both the level of expression and the number of expressing cells increases with the concentration (see Rebuttal Figure 2).

Rebuttal Figure 2. Dose response of the pAGA1 and pFIG1 induction. (A) same graphs as the ones presented in Expanded View 3 where all the cells in the population are used to calculate the expression output. (B) Same experiment as in (A) but showing the expression output only for the responding cells

6-*The authors comment on the steepness of the dose responses. To be more accurate, it would be very useful if they provide a measure of steepness, such as hill coefficient. Again, I suppose the resulting measure will be affected if non-responders are or not included.*

To follow up on the reviewer's suggestion, we have fitted the dose-response curves with a Hill function. The coefficients from the fit have been added in a small table in the Appendix Figure S9 and we discuss these new results in the main text.

7-*How do the authors explain that for AGA1 there is almost no difference between WT and bar1-delta? FIG1 shows the expected large difference.*

The reviewer should note the significant increase in the number of responding cells for both pAGA1 and pFIG1 (Expanded View 3C). In addition, we want to highlight the fact that the pAGA1 and the pFIG1 dPSTRs for the *bar1*Δ and the WT dose-responses are measured in the same cells.

8-*It is a pity the authors did most of the measurements in BAR1+ cells. The effect of Bar1 is not relevant as to the sensitivity of each promoter, and due to the effect of Bar1 in late promoters, we are left unable to compare their relative sensitivity to pheromone. As far as it is believed in the field, the only action of Bar1 is to destroy pheromone. So, doing the experiments in this way complicates the interpretation of the results.*

In retrospect, we agree with the reviewer that the characterization of the promoters in the exogenous stimulation experiments should have been done in *bar1*Δ strains. However, most experiments are performed at saturating levels of pheromone and the experiment lasts at most 1h30. Under these conditions Bar1 has only a small effect on the mating response and the dynamics and level of gene expression as shown in Appendix Figure S13. In addition, using the the BAR1+ strain for the exogenous stimulation experiments allows for a better comparison with the mating assays experiments.

Reviewer #3:

In this manuscript, Aymoz et al. studied the dynamics of the yeast molecular response to mating pheromone, using reporter systems that they previously developed. These systems (called dPSTR) are based on the relocalization of a fluorescent reporter and they offer high temporal resolution and allow to track individual live cells over time. In the present study, they monitored the transcriptional induction of 14 mating genes after exposure to pheromone and they report different kinetics of activation, with early (eg pAGA1), middle(eg pFUS2) and later promoters (eg pFIG1) being separated by up to 30 minutes in time. They controlled that this actually corresponded to delayed transcriptional activations and not technical artefacts, and that similar delays occurred in cells undergoing physiological mating (and not just in cells responding to artificial addition of pheromone). They conducted a genetic study of these delays, by either manipulating binding sites in the promoters, or by studying the response in null mutants of putative transcriptional regulators. This led to the conclusion that KAR4 acts as a relay between the early activation of AGA1 and the late activation of FIG1. Chromatin IP and MNase-protection assays demonstrated the progressive remodelling at promoters.

This is an impressive study that illustrates how a global response is in fact structured (sequential) in time. Experiments are very precise (in time and in quantification), the authors provide data that is abundant (many cells), accompanied by important controls (Northern blots, biochemical assays) and they verified the biological relevance of the different timings (mating assays). This detailed temporal description of the mating response will be interesting for the specialized readership but

also as an illustration of eukaryotic signaling responses in general. It also shows the power of the dPSTR reporters that many other labs may want to use.

Major point

The authors may have sub-exploited the wealth of data they produced with the two-colors experiments. For each cell, a time-course quantification is available for 2 promoters, but the figures seem to report only the correlation across identical time points, or across the entire data. Have the authors explored cross-correlation functions? For any pair of promoters they studied (pAGA1, pX) or (pFIG1, pKAR3), they could generate a population of such functions (one per cell). Each function may be imprecise but, together, cross-correlation functions may peak at a critical time delay. If they do, this would correlate early AGA1 signals with later promoter induction in a deterministic manner, and estimate the corresponding delay. If they do not, this would support a stochastic secondary activation of the late gene. Same is true for (Ste7-SKARS, pAGA1) and (Ste7-SKARS, pFIG1) data.

Rebuttal Figure 3 presents the cross-correlation between pAGA1-dPSTR-Y and pAGA1-dPSTR-R or pFIG1-dPSTR-R. Based on our knowledge of these two promoters, we would expect that the pAGA1/pAGA1 cross-correlation is centered around zero, while the pFIG1/pAGA1 cross-correlation should peak around 20-30 minutes. However, we see that both curves in Rebuttal Figure 3 are centered around 0 and that the pFIG1/pAGA1 cross-correlation is only slightly skewed towards higher time points. We conclude that unfortunately these type of analyses cannot be performed with our dataset. We can offer two explanations for this, the first one is that we have relatively few time points in the data set (20). In addition, the single cell traces are noisy and this noise is correlated between the two curves because it arises mainly from the segmentation process.

Rebuttal Figure 3. Cross-correlation between the nuclear enrichment of a dPSTR-R and a dPSTR-Y in the same cell.

Minor points

- Would promoter swapping of the FIG1 and AGA1 gene have any effect on mating efficiency? The manuscript is already very dense and this could be a later follow-up characterization. However, if generating strains and running a competitive mating assay is do-able, this would evaluate the biological importance of a proper sequence of events.

This is a very interesting point raised by the reviewer. We have tried to perform some experiments along these lines without much success. We have tried to place *FUS1* under the control of the *FIG1* promoter but this did not result in a particular phenotype. We

believe that the dynamics of gene expression is important for the mating process, but we have to identify the correct experimental setting where we can test this.

- *The text is a mixture of active mode "We quantified... We used.." and passive mode "Promoters were compared... were characterized...". The authors should choose one and stay consistent throughout the manuscript. Same comment about the use of present and past tense.*

After discussion with the editor, we decided to leave this as it is now. We have used the past tense to describe experiments or analysis that we have performed during the course of the study ("a dynamic protein expression reporter pFIG1-dPSTR-R was integrated"). We use the present tense to describe results ("the resulting pFIG1 expression occurs 30 minutes later")

- *Colors for times 0 and 60 min are hard to distinguish on plots 1G, 2D-E, S13-S14, and it would be informative to see a subset of time trajectories linking points of the same cells. For example on 1G, are the few dots FIG1+/AGA- from the same cells? Are they technical errors or are there few cells violating the general rule ?*

The few dots in Figure 1G are dead cells that do not respond to the stimulus. We have typically removed them from the analysis based on their high level of auto-fluorescence but some of these cells still escaped our quality control.

- *The differential kinetics of mRNA increase (Fig S7E or S7C-D) should appear in main Fig. 1 (space is available).*

The Northern blot has been moved to Figure 1.

- *Intro: "in embryonic development": sounds very restrictive for a first sentence. Immune response, tissue repairs... are also concerned.*

We have rephrased the introduction to broaden its scope.

- *Writing "temporal gradient of pheromone" (p7) is imprecise and unnecessary. The data of bar1-cells show the importance of pheromone concentration, but not a temporal gradient.*

We believe that the temporal increase in pheromone concentration is a key parameter to trigger timely late gene expression. Our exogenous stimulation experiments demonstrate that when a step increase in concentration is applied, the cells induce pFIG1 with a 40-minute delay. If this delay was also present during the mating process, the late genes would not be expressed in time to contribute to the fusion and karyogamy. Therefore, we believe that a gradual increase in pheromone concentration prepares the cells for mating and allows a rapid activation of the late genes precisely when they are needed for the fusion of the two partner cells. We have edited the text to explain this hypothesis more clearly.

- *Instead of 4 panels of supp fig 10, it may be clearer to show a dot plot with y = response time (values of x-axis of 2A) and x = basal EC in %.*

We have added a fifth panel to Appendix Figure S8 as suggested by the reviewer.

Other comments

- Intro: Two or three sentences reminding the SKARS and dPSTR systems could be added before paragraph "In this study...".
- p2: sentence "Analysis of ... still results in a delayed and broad..." is badly written.
- p2: signaling competent cells -> signaling-competent cells.
- p3: "the population moves upwards" has no biological meaning. Simply say there is a first shift along x and a later shift along y.
- p3: simultaneous activation of all mating genes -> simultaneous transcriptional activation of all mating genes
- p5: gnc5/SAGA is not strictly a remodeler. Change "nucleosome remodeling" by "nucleosome modification"
- p5: -1 histone -> -1 nucleosome.
- p6: "turned out to be" -> "was"
- p6: "accelerated the induction of the dPSTR" -> "accelerated the induction response" of the promoter
- Ref 19 is incorrectly referenced (NRG and not NPG).
- Supp Fig 5: I suggest to change the figure title "The kinetics differences between promoters is not an artifact due to the dPSTR FP and SynZip pair." by "Consistent differential kinetics between promoters after inverting dPSTR pairs."
- Legend supp Fig 9: Signtest -> Sign test.
- Colors supp fig 10D are hard to distinguish in the legend: put names next to lines?

Thank you for your suggestions and corrections. The manuscript has been updated to integrate these changes.

2nd Editorial Decision

15 January 2018

Thank you again for submitting your revised work to Molecular Systems Biology. We are now globally satisfied with the modifications made and I am pleased to inform you that we will be able to accept your paper for publication in Molecular Systems Biology pending the following minor amendments:

- Please upload individual files for each high-resolution figure and EV figures.
- Remove the paragraph called 'Supplementary Materials' on p 13 in the manuscript.

Supplementary Material

- Please rename the file from 'Supplementary Material' to 'Appendix'
- Appendix figure S4 appears twice (p.5 & 6). Please remove the duplicate page.
- Please rename the tables in the Appendix from 'Table EV[n]' to 'Appendix Table S[n]' and update the callouts in the manuscript accordingly/OR upload the EV tables, including their legends, as individual files.

Datasets

- The file "Figure Source Data" should be renamed "Dataset EV1" and explicitly called out from the main text. Please include a text-only README file that explains the content and labeling of the individual files included in the zip archive.
- Please add a Data availability section and list the doi of the imaging dataset in IDR.

Callouts

- All panels in the main figures should be cited in the text. Please add an explicit callout to Figure 1H.
- Please update the callout on page 9 from 'Sup. Figure 20' to 'Appendix Figure S20'.

Figures

- You may consider adding a scale bar to fig EV4D.

2nd Revision - authors' response

15 February 2018

Please find enclosed our revised manuscript entitled "Timing of gene expression in a cell-fate decision system" (MSB-17-8024R) for publication in Molecular Systems Biology. We corrected the manuscript following your editorial suggestions.

- The Supplementary Material is now called Appendix
- One version of the duplicated Appendix Figure S4 has been removed
- The tables have been labeled Appendix Figure S1 to S4 -We have now included two datasets: One for the single cell microscopy data and one for the biochemical experiments. They are saved in two different ZIP archives also containing a ReadMe.txt file.
- The callouts to the figures and tables have been corrected.

The last remaining item is the attribution of a DIO form the IDR repository. The raw data have been sent a couple of weeks ago along with a description of the files. We are in communication with the IDR team to provide a proper documentation of our images and analysis files.

Thank you very much for your consideration, and we are looking forward to hearing from you.

Corresponding Author Name: Serge Pelet
Manuscript Number: MSB-17-8024